# Mitochondrial Redox Balance of Fibroblasts Exposed to Ti-6Al-4V Microplates Subjected to Different Types of Anodizing

**DOI:** 10.3390/ijms241612896

**Published:** 2023-08-17

**Authors:** Anna Zalewska, Bożena Antonowicz, Julita Szulimowska, Izabela Zieniewska-Siemieńczuk, Barbara Leśniewska, Jan Borys, Sara Zięba, Paula Kostecka-Sochoń, Małgorzata Żendzian-Piotrowska, Roberto Lo Giudice, Giusseppe Lo Giudice, Piotr Żukowski, Mateusz Maciejczyk

**Affiliations:** 1Conservative Dentistry Department, Medical University in Bialystok, 15-278 Bialystok, Poland; julita.szulimowska@umb.edu.pl (J.S.); izabela.zieniewska-siemienczuk@umb.edu.pl (I.Z.-S.); paula.kostecka@umb.edu.pl (P.K.-S.); 2Dental Surgery Department, Medical University in Bialystok, 15-278 Bialystok, Poland; bozena.antonowicz@umb.edu.pl; 3Department of Analytical and Inorganic Chemistry, University in Bialystok, 15-328 Bialystok, Poland; blesniew@uwb.edu.pl; 4Department of Maxillofacial Surgery, Medical University in Bialystok, 15-278 Bialystok, Poland; jan.borys@umb.edu.pl; 5PhD School, Medical University in Bialystok, 15-278 Bialystok, Poland; 6Department of Hygiene, Epidemiology and Ergonomics, Medical University in Bialystok, 15-278 Bialystok, Poland; malgorzata.zendzian-piotrowska@umb.edu.pl (M.Ż.-P.); mat.maciejczyk@gmail.com (M.M.); 7Department of Human Pathology of the Adult and Evolutive Age G. Barresi, Messina University, 98100 Messina, Italy; roberto.logiudice@unime.it; 8Department of Biomedical and Dental Sciences and Morphofunctional Imaging, Messina University, 98100 Messina, Italy; logiudiceg@unime.it; 9Restorative Dentistry Department, Croydon University, London CR9 1DX, UK; piotr.zukowski@nhs.net

**Keywords:** antioxidants, mitochondrial redox balance, titanium implants

## Abstract

Despite the high biocompatibility of titanium and its alloys, the need to remove titanium implants is increasingly being debated due to the potential for adverse effects associated with long-term retention. Therefore, new solutions are being sought to enhance the biocompatibility of titanium implants. One of them is to increase the thickness of the passive layer of the implant made of titanium dioxide. We were the first to evaluate the effect of hard-anodized (type II) Ti-6Al-4V alloy discs on the cytotoxicity, mitochondrial function, and redox balance of fibroblasts mitochondria compared to standard-anodized (type III) and non-anodized discs. The study used fibroblasts obtained from human gingival tissue. The test discs were applied to the bottom of 12-well plates. Cells were cultured for 24 h and 7, 14, and 21 days and mitochondria were isolated. We demonstrated the occurrence of oxidative stress in the mitochondria of fibroblasts of all tested groups, regardless of the presence and type of anodization. Type II anodization prevented changes in complex II activity (vs. control). The lowest degree of citrate synthase inhibition occurred in mitochondria exposed to titanium discs with type II anodization. In the last phase of culture, the presence of type II anodization reduced the degree of cytochrome c oxidase inhibition compared to the other tests groups and the control group, and prevented apoptosis. Throughout the experiment, the release of titanium, aluminium, and vanadium ions from titanium discs with a hard-anodized passive layer was higher than from the other titanium discs, but decreased with time. The obtained results proved the existence of dysfunction and redox imbalance in the mitochondria of fibroblasts exposed to hard-anodized titanium discs, suggesting the need to search for new materials perhaps biodegradable in tissues of the human body.

## 1. Introduction

The standard treatment for mandibular fractures is surgery with titanium plates and screws. The use of osteosynthesis allows for shorter intermaxillary fixation times and faster restoration of stomatognathic function. Despite the high biocompatibility of titanium and its alloys, the necessity of removing titanium implants is increasingly debated due to the possibility of distant side effects associated with leaving biomaterials in the body for a long time. As indicated in the literature, some patients with titanium implants made of anodized alloys have been observed to have metallosis as a standard phenomenon [1,2,3,4]. It involves the deposition of metallic particles in the tissues, which are released from the implant due to friction and electrochemical corrosion [3,4,5]. It is proven, that titanium implant’s aluminum (Al) and vanadium (V) constituents have very cytotoxic effect on cells and can be released for a long time from implantation, resulting in harmful biological effects [6]. Both the presence of the titanium implant and the products of its wear can cause an immune-inflammatory response of the body [3,4,7]. Moreover, the released metal particles increase the production of free radicals (ROS) and lead to a redox imbalance in favour of oxidation reactions. Physiologically, a living organism maintains a balance between the production and inactivation of ROS, which is called redox balance. The generation of ROS in cells using oxygen as an energy source is coupled with the existence of protective systems against their action, the so-called antioxidant barrier of the body. One of the key antioxidant systems of cells is the superoxide dismutase (SOD), catalase (CAT), and glutathione peroxidase (GPx) system. GPx, along with catalase, neutralizes H_2_O_2_ formed in a dismutation reaction involving SOD. The dismutation reaction is the dismutation reaction of superoxide anion (O_2_^•−^ to H_2_O_2_. This includes NADPH oxidase (NOX), which generates large amounts of O_2_^•−^, and, sometimes, H_2_O_2_ [8]. In a study by Borys [2], a significant reduction in the activity of antioxidant enzymes is observed in the periosteum-covering titanium obturators constructed of Ti6A14V alloy, with a passive layer anodized in a standard manner, compared to the control group. This fact, according to the authors, is due to the consumption of enzymes in the process of neutralizing ROS; since the periosteum was taken between 12 and 30 months after implantation of the implant, which significantly prolonged the possibility of corrosion processes and metallosis. Depletion of antioxidant resources result in an increase in the concentrations of oxidative products of bio-particle modification, following the above-mentioned implants [2]. Evidence shows that oxidation of bio-particles interferes with the body’s regenerative processes, which, on the one hand, can lead to the need for further surgical procedures and, on the other hand, be the cause of complications in distant organs. Wear products of titanium implants were found in lymph nodes [9], brain [10], liver, and kidney [11]. Evaluation of serum and urine metal concentrations is even used as a biomarker of implant wear [12].

In the case of biomaterials such as titanium alloys used in bone reconstruction, the promotion of osteogenic differentiation and reduction of inflammation and, thus, oxidative stress (OS) are desirable. That is why researchers are looking for various solutions to increase the biocompatibility of titanium implants. Mieszkowska A. et al. [13] used fibrillar coatings with phloroglucinol (PG). These studies have shown that this type of coating significantly reduces the inflammatory response as well as promotes osteogenic differentiation. In the studies of Norris K. et al. [14], collagen fibril sheaths enriched with HE800 exopolysaccharides and GY785 EPS heteropolysaccharide derivatives were deposited on a titanium alloy (Ti6Al4V). These envelopes did not significantly alter the adhesion, morphology, or viability of osteoblast-like cells. In these studies, the influence of coatings on the behaviour of OS parameters was not assessed.

The introduction of titanium implants with a thickened titanium dioxide layer could reduce the harmful influence of titanium discs on redox balance and, consequently, on implant disintegration. This layer is expected to protect, to a large extent, against the effects of mechanical friction, and is expected to minimize the risk of ion migration from the implant alloy into the surrounding tissues. Given the lack of studies on the effect of hard anodized implants on cytotoxicity, or OS phenomenon, it is very reasonable to evaluate the effect of titanium discs made of Ti-6AI-4V alloy subjected to hard anodization (type II) on cytotoxicity, function, and redox balance of mitochondria in a fibroblast research model (ATCC-PCS-201-018) and to compare the results with those obtained in cultures of fibroblasts exposed to titanium discs constructed of Ti-6Al-4V alloy subjected to standard anodization (type III) and to titanium discs constructed of Ti-GAl-4V alloy not subjected to anodization, the so-called “raw discs”.

## 2. Results

There were no significant differences in the values of the parameters studied between the groups after 24 h of exposure, so the figures and description were omitted.

Fibroblast viability scores in the MTT assay after 14 and 21 days were significantly higher for V(st) (*p* = 0.004 and *p* = 0.0004, respectively) and V(t) alloys (*p* = 0.007 and *p* = 0.005, respectively) compared to V alloy (Figure 1).

### 2.1. Antioxidant Enzymes and Proteins

SOD activity in the mitochondria of fibroblasts exposed to titanium–vanadium discs after 7 days’ exposure was significantly higher for V(st) and V(t) alloys compared to control (*p* < 0.0001 and *p* < 0.0001, respectively), as well as compared to V alloy (*p* < 0.0001 and *p* = 0.0008, respectively). After 14 days of the experiment, SOD activity in the mitochondria of fibroblasts exposed to titanium–vanadium discs was significantly higher for V(st) and V(t) alloys compared to controls (*p* < 0.0001 and *p* = 0.007, respectively). SOD activity in the mitochondria of fibroblasts exposed to V(st) discs was significantly higher for V and V(t) alloys compared to control (*p* < 0.0001 and *p* = 0.002, respectively). SOD activity in mitochondria of fibroblasts exposed to titanium–vanadium discs after 21 days’ exposure was significantly higher for alloy V(st) compared to control (*p* < 0.0001). SOD activity in the mitochondria of fibroblasts exposed to V(st) discs was significantly higher compared to alloys V and V(t) (*p* < 0.0001 and *p* = 0.0002, respectively) (Figure 2).

CAT activity in the mitochondria of fibroblasts exposed to titanium–vanadium discs after 14 days of exposure was significantly higher for alloy V compared to control (*p* < 0.0001), as well as compared to alloys V(st) and V(t) (*p* < 0.0001 and *p* < 0.0001, respectively).

After 21 days of exposure, CAT activity in mitochondria of fibroblasts exposed to titanium–vanadium discs was significantly higher for V, V(st), and V(t) alloys compared to controls (*p* < 0.0001, *p* = 0.02, and *p* = 0.02, respectively). The activity of this enzyme in the mitochondria of fibroblasts exposed to titanium–vanadium discs was significantly higher for alloy V compared to alloys V(st) and V(t) (*p* = 0.0007 and *p* = 0.0005, respectively) (Figure 2).

GPx activity in the mitochondria of fibroblasts exposed to titanium–vanadium discs after 7 days of exposure was significantly higher for V (*p* < 0.001), V(st) (*p* < 0.0001), and V(t) (*p* < 0.0001) alloys compared to controls. GPx activity was significantly higher for alloy V(st) compared to V(t) (*p* = 0.048) (Figure 2).

The concentration of glutathione (GSH) in the mitochondria of fibroblasts exposed to titanium–vanadium discs after 7 days of exposure was significantly higher for alloys V(st) and V(t) compared to control (*p* = 0.0006 and *p* < 0.0001, respectively) and to alloy V (*p* = 0.002 and *p* < 0.0001, respectively).

The concentration of GSH in the mitochondria of fibroblasts exposed to titanium–vanadium discs after 14 days of exposure was significantly higher for alloy V(t) compared to control and alloys V and V(st) (*p* = 0.0007; *p* = 0.0004 and *p* = 0.04, respectively).

The concentration of GSH in the mitochondria of fibroblasts exposed to titanium–vanadium discs after 21 days of exposure was significantly lower for alloy V compared to control, as well as compared to the concentration of GSH in the mitochondria of fibroblasts exposed to alloys V(st) and V(t) (*p* = 0.048, *p* < 0.0001, and *p* < 0.0001, respectively). The concentration of GSH in the mitochondria of fibroblasts exposed to titanium–vanadium discs after 21 days of exposure was significantly higher for V(st) and V(t) alloys compared to controls (*p* < 0.0001 and *p* < 0.0001, respectively) (Figure 3).

### 2.2. Oxidation Products

The concentration of 4-hydroxynonenal adducts (4-HNE adducts) in the mitochondria of fibroblasts exposed to titanium–vanadium discs after 7 days of exposure was significantly higher for alloy V compared to control and alloy V(t) (*p* = 0.0003 and *p* = 0.008, respectively), moreover, the concentration of 4-HNE was higher for alloy V(st) compared to control (*p* = 0.03). The concentration of 4-HINE adducts in the mitochondria of fibroblasts exposed to titanium–vanadium discs after 14 days of exposure was significantly higher for alloy V compared to control and alloys V(t) and V(st) (*p* < 0.0001, *p* = 0.0002, and *p* < 0.0001, respectively), moreover, the concentration of 4-HNE was higher for alloy V(st) compared to control (*p* = 0.002). The concentration of 4-HNE adducts in the mitochondria of fibroblasts exposed to titanium–vanadium discs after 21 days of exposure was significantly higher for alloy V compared to control and alloys V(t), V(st) (*p* < 0.0001, *p* < 0.0001, and *p* = 0.0003, respectively) (Figure 4).

The concentration of disulfide groups in the mitochondria of fibroblasts exposed to titanium–vanadium discs after 7 days of exposure was significantly lower for alloy V and V(t), compared to the control (*p* = 0.0003 and *p* = 0.005, respectively). The concentration of disulfide groups in the mitochondria of fibroblasts exposed to titanium–vanadium discs after 7 days of exposure was significantly lower for alloy V compared to alloy V(st) (*p* = 0.03).

The concentration of disulfide groups in the mitochondria of fibroblasts exposed to titanium–vanadium discs after 14 and 21 days of exposure was significantly lower for alloy V (*p* < 0.0001 and *p* < 0.0001, respectively), V(st) (*p* < 0.0001 and *p* = 0.045, respectively), and V(t) (*p* = 0.0002 and *p* = 0.007, respectively) relative to controls. The concentration of disulfide groups in the mitochondria of fibroblasts exposed to titanium–vanadium discs after 14 and 21 days of exposure was significantly lower for alloy V, relative to alloy V(t) (*p* = 0.04 and *p* = 0.009, respectively), moreover, the concentration of disulfide groups in the mitochondria of fibroblasts exposed to titanium–vanadium discs after 21 days of exposure was significantly higher for alloy V(st) compared to alloy V (*p* = 0.001) (Figure 4).

Advanced protein glycation end products (AGE) levels in the mitochondria of fibroblasts exposed to titanium–vanadium discs after 21 days of exposure were significantly higher for V, V(st), and V(t) alloys compared to controls (*p* = 0.005, *p* = 0.005, and *p* = 0.01, respectively) (Figure 4).

### 2.3. NOX Activity

NOX activity in the mitochondria of fibroblasts exposed to titanium–vanadium discs after 7 days of exposure was significantly higher for V, V(st), and V(t) alloys compared to controls (*p* = 0.0004, *p* = 0.02, and *p* = 0.01, respectively).

NOX activity in the mitochondria of fibroblasts exposed to titanium–vanadium discs after 14 days of exposure was significantly higher for alloy V relative to the control (*p* < 0.0001), as well as relative to the activity of this enzyme in the mitochondria of fibroblasts exposed to alloy V(t) (*p* = 0.03). NOX activity in the mitochondria of fibroblasts exposed to titanium–vanadium discs after 14 days of exposure was significantly higher for V(st) alloy relative to control (*p* = 0.02).

NOX activity in the mitochondria of fibroblasts exposed to titanium–vanadium discs after 21 days of exposure was significantly higher for V, V(st), and V(t) alloys relative to controls (*p* < 0.0001, *p* = 0.005, and *p* = 0.03, respectively). NOX activity in the mitochondria of fibroblasts exposed to titanium–vanadium discs after 21 days of exposure was significantly higher for alloy V relative to V(t) (*p* = 0.013) (Figure 5).

### 2.4. Apoptosis

Caspase -3 (CASP-3) activity in the mitochondria of fibroblasts exposed to titanium–vanadium discs after 14 days of exposure was significantly higher for V, V(st), and V(t) alloys relative to controls (*p* < 0.0001, *p* < 0.0001, and *p* < 0.0001, respectively). CASP-3 activity in the mitochondria of fibroblasts exposed to titanium–vanadium discs after 21 days of exposure was significantly higher for V and V(st) alloys relative to controls (*p* < 0.0001 and *p* = 0.001, respectively) and relative to CASP-3 activity in the mitochondria of fibroblasts exposed to titanium–vanadium discs from V(t) alloy (*p* = 0.002 and *p* = 0.045, respectively) (Figure 5).

### 2.5. Nitrosative Stress

The concentration of peroxynitrite (ONOO-) in the mitochondria of fibroblasts exposed to titanium–vanadium discs after 21 days of exposure was significantly higher for V, V(st), and V(t) alloys compared to controls (*p* = 0.005, *p* = 0.003, and *p* = 0.007, respectively) (Figure 6).

The concentration of 3-nitrotyrosine (3-NT) in the mitochondria of fibroblasts exposed to titanium–vanadium discs after 14 days of exposure was significantly higher for V and V(st) alloys compared to controls (*p* = 0.0005 and *p* = 0.02, respectively).

The concentration of 3-NT in the mitochondria of fibroblasts exposed to titanium–vanadium discs after 21 days of exposure was significantly higher for alloys V, V(st), and V(t) relative to controls (*p* < 0.0001, *p* < 0.0001, and *p* = 0.002, respectively). The concentration of 3-NT in the mitochondria of fibroblasts exposed to titanium–vanadium discs after 21 days of exposure was significantly higher for V and V(st) alloys compared to V(t) alloy (*p* < 0.0001 and *p* = 0.04, respectively); it was also significantly higher for V alloy compared to V(st) (*p* = 0.008) (Figure 6).

### 2.6. Mitochondrial Complexes

The activity of complex I in the mitochondria of fibroblasts exposed to titanium–vanadium discs on the 7th day of the experiment was significantly lower for alloys V, V(st), and V(t) compared to the control (*p* < 0.0001, *p* < 0.0001, and *p* < 0.0001, respectively). After 7 days of exposure, complex I activity was significantly higher for alloys V(st) and V(t) compared to alloy V (*p* = 0.0001 and *p* = 0.006, respectively). Complex I activity in the mitochondria of fibroblasts exposed to titanium–vanadium discs on day 14 of the experiment was significantly lower for alloys V and V(st) compared to controls (*p* < 0.0001 and *p* < 0.0001, respectively). It was also significantly higher for alloy V(t) compared to alloys V(st) and V (*p* = 0.003 and *p* < 0.0001, respectively). Complex I activity in the mitochondria of fibroblasts exposed to titanium–vanadium discs on day 21 of the experiment was significantly lower for alloy V compared to control and alloys V(st) and V(t) (*p* < 0.0001, *p* = 0.0002, and *p* = 0.01, respectively) (Figure 7).

Complex II activity in the mitochondria of fibroblasts exposed to titanium discs after 7 days of exposure was significantly lower for V and V(st) alloys compared to controls (*p* < 0.0001 and *p* = 0.03, respectively). The activity of complex II in the mitochondria of fibroblasts exposed to titanium discs with vanadium after 7 days of exposure was significantly lower for alloy V compared to alloys V(st) (*p* = 0.025) and V(t) (*p* < 0.0001), as well as a significantly higher activity of the complex in question was noted for alloy V(t) compared to V(st) (*p* = 0.03). The activity of complex II in the mitochondria of fibroblasts exposed to titanium–vanadium discs after 14 days of exposure was significantly lower for V, V(st), and V(t) alloys compared to controls (*p* < 0.0001, *p* < 0.0001, and *p* = 0.002, respectively). Complex II activity in the mitochondria of fibroblasts exposed to titanium–vanadium discs after 14 days of exposure was also significantly higher for V(st) and V(t) alloys compared to V (*p* = 0.03 and *p* = 0.0003, respectively). Complex II activity in the mitochondria of fibroblasts exposed to titanium discs after 21 days of exposure was significantly lower for V and V(st) alloys compared to controls (*p* < 0.0001 and *p* = 0.0008, respectively). Complex II activity in the mitochondria of fibroblasts exposed to titanium–vanadium discs after 21 days of exposure was also significantly higher for V(st) and V(t) alloys compared to V (*p* < 0.0001 and *p* < 0.0001, respectively) (Figure 7).

Cytochrome c oxidase (COX) activity in the mitochondria of fibroblasts exposed to titanium–vanadium discs after 7 (*p* < 0.0001, *p* < 0.0001, and *p* < 0.0001, respectively) and 14 (*p* < 0.0001, *p* < 0.0001, and *p* < 0.0001, respectively) days of exposure was significantly lower for V, V(st), and V(t) alloys compared to controls. On day 21, COX activity in mitochondria of fibroblasts exposed to titanium–vanadium discs was significantly lower only for V and V(st) alloys compared to control (*p* < 0.0001 and *p* < 0.0001, respectively). At day 7, the activity of the enzyme in question was significantly higher for V(st) alloy compared to V (*p* = 0.02) and, after 21 days, COX activity for V and V(st) alloys was significantly lower compared to COX activity in the mitochondria of fibroblasts exposed to V(t) alloy (*p* = 0.01 and *p* = 0.04, respectively) (Figure 7).

Citrate synthase (CS) activity in the mitochondria of fibroblasts exposed to titanium–vanadium discs after 7 (*p* < 0.0001, *p* < 0.0001, and *p* < 0.0001, respectively), 14 (*p* < 0.0001, *p* < 0.0001, and *p* < 0.0001, respectively), and 21 (*p* < 0.0001, *p* = 0.004, and *p* = 0.006, respectively) days of exposure was significantly lower for V, V(st), and V(t) alloys compared to controls. CS activity in the mitochondria of fibroblasts exposed to titanium–vanadium discs after 14 days of exposure was significantly higher for V(t) alloy compared to V (*p* = 0.015) and, after 21 days of exposure, CS activity in the mitochondria of V(st) and V(t) alloys was significantly higher compared to CS activity in the mitochondria of fibroblasts exposed to V alloy (*p* = 0.02 and *p* = 0.001, respectively) (Figure 7).

### 2.7. Quantitative Analysis of Metal Content in the Medium

The titanium content of the medium taken after 3 (*p* = 0.002 and *p* = 0.003, respectively), 6 (*p* = 0.02 and *p* = 0.001, respectively), 15 (*p* = 0.008 and *p* = 0.007, respectively), and 21 (*p* = 0.008 and *p* = 0.007, respectively) days of exposure of fibroblasts to titanium–vanadium discs was significantly higher for V(t) alloy compared to V and V(st) alloys. The titanium content of the medium taken after 6 (*p* = 0.04), 15 (*p* = 0.04), and 21 (*p* = 0.03) days of exposure of fibroblasts to titanium–vanadium discs was significantly higher for alloy V compared to V(st) (Table 1).

Aluminum content in the medium taken after 3 (*p* = 0.001 and *p* = 0.003, respectively) and 6 (*p* = 0.04 and *p* = 0.001, respectively) days’ exposure of fibroblasts to titanium–vanadium discs was significantly higher for V(t) alloy compared to V and V(st) alloys.

Aluminum content in the medium taken after 6 (*p* = 0.03) and 15 (*p* = 0.04) days’ exposure of fibroblasts to titanium–vanadium discs was significantly higher for alloy V compared to alloy V(st). Aluminum content in the medium taken after 15 days’ exposure of fibroblasts to titanium–vanadium discs was significantly higher for alloy V(t) compared to alloy V(st) (*p* = 0.001). The aluminum content of the medium taken after 21 days’ exposure of fibroblasts to titanium–vanadium discs was significantly higher for alloy V(t) compared to alloy V (*p* = 0.008) (Table 2).

Vanadium content in the medium taken after 3 (*p* = 0.001 and *p* = 0.001, respectively), 6 (*p* = 0.001 and *p* = 0.001, respectively), 15 (*p* = 0.0001 and *p* = 0.0001, respectively), 21 (*p* = 0.001 and *p* = 0.001, respectively) days’ exposure of fibroblasts to titanium–vanadium discs was significantly higher for alloy V(t) compared to alloys V and V(st). Vanadium content in the medium taken after 15 days’ exposure of fibroblasts to titanium–vanadium discs was significantly higher for alloy V compared to alloy V(st) (*p* = 0.02) (Table 3).

### 2.8. Concentrations of Growth Factors in the Medium

The concentration of fibroblast growth factor 2 (FGF 2) in the medium collected after 7, 14, and 21 days’ exposure of fibroblasts to titanium–vanadium discs was significantly lower for V alloys (*p* = 0.004, *p* < 0.0001, and *p* < 0.0001), V(st) (*p* = 0.03, *p* < 0.0001, and *p* < 0.001, respectively) and V(t) (*p* < 0.05, *p* < 0.0001, and *p* = 0.0004, respectively) compared to controls (Figure 8).

The concentration of vascular endothelial growth factor A (VEGF-A) in the medium collected after 7, 14, and 21 days’ exposure of fibroblasts to titanium–vanadium discs was significantly lower for V alloys (*p* < 0.0001, *p* < 0.0001, and *p* < 0.0001, respectively), V(st) (*p* < 0.0001, *p* < 0.0001, and *p* < 0.0001, respectively), and V(t) (*p* < 0.0001, *p* < 0.0001, and *p* < 0.0001, respectively) compared to control (Figure 8).

### 2.9. Correlations

Day 7 showed a negative correlation between peroxynitrite concentration and complex I activity in the mitochondria of fibroblasts treated with all three titanium discs (r = −0.61; r = −0.77; r = −0.75; *p* < 0.05).

## 3. Discussion

The standard treatment for craniofacial trauma, facial defects, and tumours is surgery, based on the use of plates and screws constructed of titanium alloys. Despite the claimed biocompatibility of titanium, including its alloys, by its manufacturers, there are increasing reports of the need to remove titanium implants, due to the distant side effects observed by many investigators [2,3,7,15,16]. The surface of titanium implants is covered with a passive layer of TiO_2_, formed by a process called standard anodization. Although the passive layer was intended to reduce the corrosion potential of the alloy, many patients experience corrosion of the alloy and deposition of metallic particles at the implant site [3,17,18]. It is worth noting that the described phenomenon is particularly common in implants that are subjected to high forces and stresses, as is the case in the mandible.

The mere presence of a titanium implant, let alone the products of its wear, disrupts the body’s immune processes and causes OS and inflammation, which can lead to the need for further surgery and result in complications in distant organs [9,10,11]. In order to reduce side effects in the form of inflammation and OS, implants with a thickened TiO_2_ layer were created, formed by type II anodization. This layer is expected to protect the alloy from the effects of mechanical friction and, in addition, to minimize the risk of ion migration from the alloy into the surrounding tissues, presumably reducing the risk of developing OS and inflammation. There are no reports in the literature evaluating the effect of titanium implants coated with a TiO_2_ layer, formed by the type II anodization process, on the redox balance of the implanted tissues, as well as their comparison with titanium implants anodized in a standard (type III) manner and without a passive layer on the surface.

For ethical reasons, the current study was conducted on cell culture of primary human gingival fibroblasts (Human Primary Gingival Fibroblasts, ATCC-PCS-201-018). It should be emphasized, the results obtained in this way cannot be translated directly to humans. The fact is that in studies evaluating titanium implants, mainly two types of cells are used, osteoblasts and fibroblasts, as both types show high adhesion to implants. As studies have shown, osteoblasts prefer rougher surfaces, while fibroblasts prefer smooth surfaces [19,20]. Since the titanium discs used had a smooth, polished surface, the latter cell type was used for the study. Based on the literature, the details of the handling of the titanium discs were also established already during the cell culture process (the degree of confluence of the monolayer at the time of “implantation”, the times of cell culture, and how the titanium discs were located in the cell culture wells). The titanium discs were used once, and the entire cell culture process was carried out by one person (I.Z-S.).

In the study, titanium–vanadium alloy (Ti4AI16V) was used in three surface layer configurations: an alloy without a TiO_2_ layer and two alloys with a passive layer: one standard anodized, the other with type II anodization on the implant surface. The metabolic activity of fibroblasts after exposure to titanium implants was evaluated. An MTT assay was performed after 24 h and 7, 14, and 21 days of culture with titanium discs. Differences in cell viability were observed only on the 14th and 21st day of the experiment; the metabolic activity of fibroblasts incubated with the disc devoid of the passive layer significantly decreased (the result is presented as % of control) compared to the other two test groups. The metabolic activity of fibroblasts exposed to discs with an anodized surface did not differ according to the type of anodization. Similarly, greater cytotoxicity of implants lacking the TiO_2_ layer compared to standard anodized implants was demonstrated by El-Shenawy N. et al. [21]. According to Tsaryk et al. [15], reduced metabolic activity of fibroblasts may be a consequence of growth inhibition, apoptosis, or necrosis as a result of an increase in unneutralized oxygen and nitrogen free radicals in cells exposed to a standard anodized implant.

TiO_2_ layer on the surface of the implant is believed to be a protective layer and determines the biocompatibility of titanium implants. Bikondoa et al. [22] showed, however, that defects such as oxygen vacancies in the model TiO_2_ surface, mediate water dissociation. Evidence of the reactivity of titanium alloy is also evidenced by elevated titanium concentrations in the serum and distant organs of patients after implantation, as we have previously written. It is believed that the release of titanium may be the result of an anodic corrosion process, where the TiO_2_ layer is damaged. The cathodic part of the corrosion process involves oxygen reduction at physiological pH with the generation of ROS and non-radical compounds, such as hydrogen peroxide H_2_O_2_ [5,23,24]. Lin et al. [25] claim that the thickening of the TiO_2_ layer in biological solutions is due to the continuous corrosion process of titanium implants in the human body. Wear particles formed at the bone/implant interface can disturb the protective TiO_2_ layers, which further induces corrosion, leading to a synergistic fretting-wear process. In addition to the possible formation of ROS by the titanium alloy itself as a result of cathodic corrosion, titanium can be exposed to ROS produced by inflammatory cells coming into contact with titanium immediately after implantation or when cultured with stimulated phagocytic cells present in the culture medium. The phenomenon of “oxygen burst” and generation of huge amounts of O_2_^•−^ and H_2_O_2_ then occurs.

One of the most important processes occurring in mitochondria is oxidative phosphorylation. This process is accompanied by the production of energy stored as ATP by the mitochondrial complex system (I–IV). About 0.1–1% of the oxygen consumed in the mitochondria is converted into superoxide anion (O_2_^−^), which undergoes a dismutation reaction catalyzed by SOD. Under physiological conditions, complexes I and III are responsible for the reduction of oxygen to O_2_^−^ [26]. The potential sites of mitochondrial ROS formation are located in complex I [27] and complex II [28], and their most intense production is due to electron flow from complex I to complex II [28]. Throughout the experiment, we observed reduced (vs. control) mitochondrial complex I activity in the mitochondria of fibroblasts exposed to titanium discs without a passive layer. We also recorded a reduction in complex I activity (vs. control) in fibroblasts exposed to hard-anodized discs, but only on day 7, and in group V(st) on days 7 and 14 of exposure. The lowest activity, among the study groups, was recorded in the group of fibroblasts exposed to titanium discs without a passive layer. On day 14, NOX activity in the group of fibroblasts exposed to hard-anodized discs (V(t)) was significantly higher vs. the V(st) group. The negative correlation observed at day 7 between peroxynitrite concentration and complex I activity in the mitochondria of fibroblasts of all three groups tested could indicate an increase in the nitration process of 3-tyrosine and inactivation of complex I subunits. Reduced activity of complex I contributes to increased generation of ROS and impaired synthesis of high-energy compounds. We also observed reduced activity of complex IV (COX, cytochrome c oxidase), closely related to energy metabolism. A decrease in the activity of the I and IV complexes might result in a drastic decrease in ATP synthesis, followed by reduced biosynthesis of collagen, elastin, and proteoglycans used in the process of tissue healing [29]. It should be emphasized that electron transport and oxidative phosphorylation are closely linked. Respiratory chain dysfunction results in a halt of oxidative phosphorylation in the mitochondrion. The energy released by electron transport is not stored as ATP, it is dissipated as heat [29]. Energy deficits in the course of exposure to titanium implants can also develop for other reasons. CS is a key enzyme involved in the functioning of the tricarboxylic acid cycle. Its reduced activity, throughout the experiment, in all test groups vs. control, may reflect inhibition of the activation of this cycle, and, thus, indicate a further decrease in ATP production [30].

As studies have shown, a reduction in mitochondrial transmembrane potential is a recognized early marker of apoptosis [31]. Deficiency of energy stored in the form of ATP could be responsible for the increased apoptosis observed from day 14 of the experiment, when Casp 3 activity in the mitochondria of fibroblasts of all study groups was elevated vs. the control group. Previous studies have shown that reduced ATP levels lead to increased apoptosis of hepatocytes [32]. In the mitochondria of fibroblasts exposed to discs with type II anodization at day 21, the process of apoptosis slowed down, and the activity of Casp 3 was at the level of this enzyme in the control group. Elevated activity of this enzyme persisted in groups of fibroblasts exposed to titanium discs without a passive layer and standard anodized discs.

Extremely important, in terms of the protective properties of the TiO_2_ passive layer formed by type II anodization, is the reported lack of disruption of complex II activity in the mitochondria of fibroblasts exposed to the disc coated with this layer vs. the control group on days 7 and 21 of the experiment. Succinate dehydrogenase (complex II) also has another role, as a structural and regulatory component of the mitochondrial ATP-sensitive K^+^ channel (mitoKATP) [33]. Increased flow of potassium ions into the mitochondrial matrix via the ATP-sensitive potassium channel has been shown to provide protection against ischemia–reperfusion injury. Ischemia–reperfusion injury is produced when blood circulation in tissue, including bone tissue, is restored after being temporarily stopped. It can result in cell damage or even cell death, which depends on the timing and severity of the blood flow disruption [34].

It can be seen that the behaviour of the antioxidant defence of fibroblasts exposed to titanium discs lacking the passive layer differs significantly from the changes in the antioxidant defence of fibroblasts exposed to anodized titanium discs, with more adverse changes seen in fibroblasts exposed to standard anodized discs. The intensification of the dismutation (↑SOD) reaction from day 7 of the experiment in the groups of fibroblasts exposed to anodized discs (V(st) and V(t)) vs. control persisted until the end of culture, which is most likely due to the increase in O_2_^−^ generation. Surprisingly, there was no difference in the activity of this enzyme between the group of fibroblasts exposed to discs without a passive layer and the control group. The lack of change in CAT activity at day 7 in all groups of fibroblasts exposed to titanium implants compared to controls, with a concomitant increase in GPx activity (in all groups of fibroblasts exposed to titanium implants compared to controls), suggests a small increase in H_2_O_2_ generation. Indeed, GPx is known to have a much higher affinity for H_2_O_2_ than CAT and its activity increases with non-significant increases in H_2_O_2_ generation [35]. However, it is unclear what magnitude defines the term “non large increase”. It should be emphasized that there were no differences in GPx activity between the study groups throughout the experiment. As of day 14 of the experiment, CAT activity increased in all study groups compared to the control group, and, as with GPx, it was highest in the group of fibroblasts exposed to implants lacking a passive layer and did not differ between groups of fibroblasts exposed to titanium discs with a standard and type II anodized passive layer. The lack of significant differences in GPx activity in all study groups from day 14 of the experiment to its conclusion suggests a large increase in H_2_O_2_ generation at which CAT becomes active. High concentrations of H_2_O_2_ inhibit GPx activity [35]. The putative high concentration of H_2_O_2_ may have both negative and positive significance. Lee et al. [36] have shown that both titanium particles and the TiO_2_ layer can interact with H_2_O_2_ leading to the formation of hydroxyl radicals, which are among the most dangerous ROS. On the other hand, it is known that high concentrations of H_2_O_2_ may be desirable in the process of indentation of a titanium implant and in wound healing in general. The generation of large amounts of H_2_O_2_ at the implant/wound site and the resulting gradient in its concentration are essential for rapid leukocyte recruitment and, thus, implant engraftment/wound healing [37].

The reduction in GSH concentration, initiated on day 7 of the experiment, in the group of fibroblasts exposed to titanium discs without a passive layer vs. the other test groups, persists throughout the experiment. In the groups of fibroblasts exposed to anodized discs, we observed an increase in GSH concentration vs. the control group and the group exposed to discs without anodization. Given that the overriding function of GSH is to maintain the thiol groups of proteins in a reduced state, its deficiency/consumption could be a reason an increased reduction of protein thiol groups in V group. However, the observed increases in GSH pool concentrations in fibroblasts exposed to titanium discs with a passive layer anodized with standard and type II anodization vs. control appear insufficient to protect protein thiol groups from oxidation (↑SS concentration in V(st) and V(t)) vs. the control). Oxidation of disulfide groups can have a very negative reflection on all cellular processes, since keeping them in a reduced state is an essential factor for the activity of enzymes, messenger proteins, or transport proteins. An oxidation of the disulfide groups of lysyl oxidase causes impaired wound healing as a result of abnormal collagen synthesis [38].

Changes in the activity of ROS-neutralizing enzymes, or changes in concentrations of small-molecule antioxidants observed in fibroblast cells, are not evidence of OS. They are, certainly, an adaptive response to the increase in ROS generation. ROS are capable of oxidizing any of the cellular components: proteins, lipids, DNA, RNA, or sugars, with a single biomarker of oxidative damage having little clinical value in diagnostic or prognostic terms. The most commonly measured indicators of oxidative modification include 4-HNE adducts with proteins, the concentration of disulfide groups, and AGE. 4-HNE adducts are simple to determine, and their concentration is directly proportional to the degree of cell membrane damage [39,40].

In the group of fibroblasts exposed to titanium discs lacking a passive layer, the concentrations of lipid peroxidation products, throughout the experiment, were higher compared to the other two groups of subjects and the control group. On days 7 and 14, elevated concentrations of 4-HNE adducts vs. control were observed in the group of fibroblasts exposed to standard anodized titanium discs. Of note is the absence of any change in the concentration of 4-HNE adducts in the group of fibroblasts exposed to passive layer discs with type II anodization vs. control. This result indicates the so-called reversibility of the process of lipid peroxidation in the situation of exposure to titanium discs with a passive layer with II type of anodization, as described in the case of salivary glands of rats exposed to a carbohydrate-rich diet [41]. Moreover, Lushchak refers to such a situation as “low-intensity OS”, which is of great clinical significance in the case of biomaterial [42,43]. Low-intensity OS can have a very beneficial effect, as it increases cell survival. This effect is associated with the activation of a number of cellular processes directed at increasing cellular resistance to OS [44]. The reversibility of lipid peroxidation is of great clinical importance. Lipid peroxidation products are considered a marker of osteoclast activity, and it is even believed that their concentration is directly proportional to osteoclast activity around titanium implants and screws [45].

An increase in AGE concentrations in all groups of fibroblasts exposed to titanium discs vs. the control group was observed at the final stage of the experiment. Because AGE can activate a number of inflammatory reactions leading to accelerated vascular damage and reduced vascular permeability [46], an increase in their concentration may be responsible for impaired angiogenesis in the wound healing region [47]. A study by Quintero et al. [48] indicates that excessive AGE formation inhibits bone turnover and weakens osteointegration and implant stability. Increased OS in the form of increased concentrations of oxidative products of modification after titanium implants constructed of Ti6A14V alloy, with a passive layer anodized as standard, was also observed in the periosteum in the vicinity of the implants, 12–30 months after surgery [2]. They also consider incorporating antioxidant prophylaxis to minimize oxidative damage at the anastomotic site. However, it should be remembered that exogenous antioxidants used for bone healing are beneficial in patients with systemic diseases such as diabetes, rheumatoid arthritis etc. The effect of these antioxidants on bone healing in healthy patients seems to be different. In the latter case, the antioxidants are believed to neutralize the levels of ROS necessary for the healing process. Physiological levels of ROS play a role of second messengers to mediate responses via redox reactions that are required for proper cell function and survival [49].

The expression of specific growth factors, such as fibroblast growth factors (FGFs), platelet-derived growth factors (PDGFs), transforming growth factor-beta (IGF-s), vascular endothelial growth factor (VEGF), and bone morphogenetic proteins (BMPs) in fibroblasts, osteoblasts, and endothelial cells during healing confirms the key role of these secreted factors in the bone repair process [50,51]. In further stages of healing, VEGF is an important factor stimulating the proliferation of endothelial cells and their progenitors, which build newly formed blood vessels on a scaffold formed from collagen and other extracellular matrix proteins. VEGF-A has also been found to play a direct role in the differentiation and maturation of osteoblasts [52]. FGF-2 as a mitotic promoter can induce mitosis and accelerate cell proliferation, CM formation, and remodelling, which promotes faster wound healing [53].

Throughout the experiment, we observed significantly lower concentrations of VEGF-A and FGF-2 in the medium taken from fibroblasts exposed to titanium discs regardless of the presence or absence of a passive layer. These results suggested that the presence of titanium implants interferes with the processes of angiogenesis and bone tissue regeneration. The observed deficiency of FGF-2 may result in a slower rate of mineralization and bone bridging in the fracture defect, as well as a reduced number of osteocytes in the bone healing after injury, as observed in the study of Murakami et al. [54] and Pilmane et al. [55].

Metal content analysis was performed in mineralized medium samples by plasma ionization mass spectrometry (ICP-MS) using an external calibration method. Ion release was not determined in the cell lysate, for technical reasons. Therefore, the percentage of these ions that could penetrate the cells is not known, so it is difficult to look for cause-and-effect relationships between the concentration of ions in the medium and the parameters studied. The release of ions from titanium discs was highest at the beginning of the experiment and decreased with time. Throughout the experiment, the concentration of titanium, aluminum, and vanadium ions were highest in the medium taken from fibroblasts exposed to titanium discs with a hard-anodized passive layer.

It is important to highlight the weaknesses of the current experiment. First, the research was conducted in a cellular model, which does not reflect the complex relationships and interactions at the implant–bone interface. Physiologically, the forces exerted on the implant material consist of tensile, compressive, and shear elements, and such conditions cannot be reproduced in the research model used. Secondly, we used only one type of cell, which does not allow us to reproduce the biological processes occurring in bone tissue. When analysing the results presented here, it should be taken into account that we also evaluated only some markers of oxidative modification and antioxidant barrier elements. Evaluation of other markers of oxidative stress may completely or partially change our observations and assumptions. We used one thickness of the anode layer; it is not known how other thicknesses of this layer would affect biological systems. It is also not known whether the release of ions would be at a similar level.

## 4. Materials and Methods

The study was approved by the Bioethics Committee of the Medical University of Bialystok (resolution no. APK.002.280.2021).

### 4.1. Cell Culture

The study used fibroblasts obtained from human gingival tissue (Human Non-Genetically Modified Primary Gingival Fibroblasts; Human Primary Gingival Fibroblasts, ATCC-PCS-201-018), which were purchased from ATCC (Manassas, VA, USA). Cells were cultured in fibroblast-dedicated basal medium (Fibroblast Basal Medium, ATCC, PCS-201-030^TM^), which was enriched with antibiotics and antimycotics (antibiotic–antimycotic, Gibco, Thermo Fisher, Waltham, MA USA 02451 15240062: penicillin (10 U/mL), streptomycin (10 µg/mL), and amphotericin B (25 ng/mL)) and a bovine serum fibroblast growth kit (ATCC, PCS-201-041^TM^: recombinant human fibroblast growth factor (rh FGF b) (5 ng/mL), L-glutamine (7.5 mM), ascorbic acid (50 ug/mL), hydrocortisone (1 µg/mL), rh insulin (5 µg/mL), bovine serum (2%), and also red phenol solution (Sigma, Gdynia, Poland, P0290-100ML, 33 µM)). Cell cultures were conducted at 37 °C in an atmosphere of 5% CO_2_ (Forma Steri-cycle i160). Cells were seeded into culture bottles (150 cm^2^ with filter, sterile, Bionovo, Legnica, Poland) as well as plates (12-well, sterile, CELLSTAR, VWR, Poland) at a density of 3000 cells per cm^2^. Cells from the 3rd passage were used for the experiments. Cell morphology was assessed daily using a Delta Optical IB-100 inverted optical microscope (catalogue number DO-3709, Delta Optical, Poland; magnification 40×, 100×, 200×). Culture medium (ATCC, PCS-201-030^TM^) was changed every 2 days. The work was carried out under aseptic conditions, in a laminar airflow chamber (Telstar Aeolus V, DanLab, Białystok, Poland). Cells were cultured in 150 cm^2^ bottles (Bionovo, Legnica, Poland) until a confluence of 80% was reached (as recommended by the manufacturer). The medium was harvested, after which the cells were washed twice with sterile PBS buffer (phosphate-buffered saline; 0.02 M, pH 7.3) (1×; PAN-Biotech GmbH, P04-36500, Aidenbach, Germany), and heated to 37 °C (in an Aquarius dry pellet bath, DanLab, Białystok, Poland). Then, cells were trypsinized with 3 mL of freshly diluted trypsin solution (1:10, *V:V*; Gibco, Thermo Fisher, Waltham, MA USA 02451 15240062, P04-36500). After a 4 min incubation, trypsin was neutralized by adding 10 mL of medium. Then, the cells were sieved into 12-well plates. Cells were cultured by adding 1 mL of medium to each well. Once the cells reached 80% confluence (average 3–4 days), titanium/polystyrene discs were applied to the bottom of the plates. All variants of the experiment were carried out in 6 independent experiments. Cells were cultured for 24 h and 7, 14, and 21 days. After sufficient time (Le., 24 h and 7, 14, and 21 days), cells were trypsinized and centrifuged (1500 rpm, 5 min, 25 °C, Centrifuge 5804 R, Poland).

Isolation of mitochondria was performed using the method described previously [56]. After collecting the medium, the cells were washed twice with ice-cold PBS (0.02 M, pH 7.3, PAN-BIOTECH, P04-36500) and then trypsinized (as described above). After centrifugation (10,000× *g*, 4 °C, 10 min), the supernatant fluid was discarded, and the cell pellet was resuspended in hypotonic buffer (5 mL per 1 g of pellet) containing 100 mM sucrose (Chempur, 427720906, Poland), 10 mM MOPS (4-morpholinopropanesulfonic acid, Sigma-Aldrich, M1254, Poland), 1 mM EGIA (ethyleneglycol-O-O’-bis(2-aminoethyl)-N,N,N′,N′-tetraacetic acid, Sigma-Aldrich, M6768, Poznań, Poland), and allowed to swell on ice for 10 min. In the next step, the samples were homogenized with a glass up/down hand homogenizer, after which hypertonic buffer (1.25 M sucrose, 10 mM MOPS) was added and diluted with isolation buffer (75 mM mannitol, 225 mM sucrose, 10 mM MOPS, 1 mM EGTA (Sigma-Aldrich, E0396, Poznań, Poland), 0.1% BSA without fatty acids (Sigma-Aldrich, A8806, Poznań, Poland)). The samples were then subjected to centrifugation (930× *g*, 4 °C, 5 min), after which the homogenization process was repeated 6 times. After another centrifugation (10,300× *g*, 4 °C, 20 min), the supernatant fluid was removed, and 5 mL of MiPO5 buffer (110 mM sucrose, 60 mM K-lactobionate (Pol-Aura, PA-03-2515-K#25G, Morąg, Poland), 20 mM HEPES (hydroxyethyl piperazine ethanesulfonic acid, Sigma-Aldrich, H4034, Poland), 10 mM KH_2_PO_4_ (Sigma-Aldrich P9791, Poznań, Poland), 3 mM MgCl_2_ × 6H_2_O 0.5 mM EGTA (Sigma-Aldrich, E0396, Poznań, Poland), 20 mM taurine (Merck, 8086160005, Warszawa, Poland), 0.1% BSA without fatty acids), and, once again, the samples were subjected to centrifugation (10,300× *g*, 4 °C, 20 min). After centrifugation, the supernatant liquid was discarded, and 50 µL of MiPO5 buffer was added to the pellet. The samples were mixed on a vortex and used immediately for the assays.

### 4.2. Titanium Discs

The discs were made to individual order by ChM sp. z o. o. (Lewickie, Poland). All discs were polished according to a standard polishing procedure to a surface roughness of Ra ≤ 0.63. Rosler Keramo-Finish Compound 629 (LM Zalewski, Chwaszczyno, Poland) was used for polishing.

The study used discs with a diameter of 21 mm and a thickness of 1 mm, constructed of titanium alloy Ti-6AI-4V containing a minimum of 88 wt% titanium, as well as 6 wt% aluminum and 4 wt% vanadium. Other titanium alloys produced by ChM do not contain vanadium. They may contain an admixture of niobium or a higher carbon content. The titanium discs were constructed in three varieties depending on the process of formation of the passive layer: standard anodized discs (III type) (V(st)), cold anodized discs (II type) (V(t)), and raw discs not subjected to the anodizing process (V). The control group consisted of polystyrene discs of the same diameter and thickness as the titanium discs.

The discs have undergone the following treatment:

1. Discs without a passive layer, raw: they have not been electrochemically treated.

2. Type III (standard) anodization: the discs were subjected to a washing process to remove the polishing paste. Then, the discs were chemically deoxidized in a solution of a mixture of HF/HNO_3_ acids, and subjected to a Type III anodizing process in a 0.5 M H_2_SO_4_ solution at voltages up to 100 V. The discs were washed again to remove residues of the electrochemical treatment bath.

3. Type II (hard anodizing): the disks were anodized in accordance with AMS 2488D (“Anodic Treatment—Titanium and Titanium Alloys Solution pH 13 or Higher”), this specification describes the technical requirements related to the production of an electrolytic conversion (anodic) coating on titanium and titanium alloys. It describes the properties of the coating. The discs were washed to remove residues such as polishing paste. The disks were then chemically deoxidized in a solution of a mixture of HF/HNO_3_ acids, and subjected to a Type Il anodizing process in a NaOH solution with a pH above 13 at voltages up to 100 V. The discs were washed again to remove residues of the electrochemical treatment bath.

The metallographic microsection of the Ti6Al4V sample was analysed using the FEI Quanta 3D FEGSEM scanning electron microscope (The Netherlands) equipped with an EDAX X-ray spectrometer.

Observations were conducted using backscattered electron (BSE) images at 2000×–10,000× magnification. Analysis of chemical composition—X-ray spectroscopy with energy dispersion. The thickness of the layer on the exterminations measured in an electron microscope was in the range of 1–2 micrometres; the average value was 1.5 +/− 0.5). It was not technically possible to prepare microsections for measuring the thickness of the standard—layer thickness was too small. The images obtained suggest that the surface topography of both types of layers was similar. Coating nanohardness tests determined by the instrumental method on the Micro Combi Tester device (Anton Paar GmbH, Ostfildern, Germany). The nanohardness value of the type II anodic layer was 160% of the value of the standard layer.

After the treatment process, the discs were rinsed in water and then fully immersed in an aqueous detergent solution (Eskaphor N6650, Haugh Chemie, Milanówek, Poland), followed by ultrasonic cleaning in an Elmasonic X-tra Pro 800 (ELMA, Warszawa, Poland) washer for 15 min at 130 kHz and a bath temperature of 55 ± 5 °C. The ultrasonic washing process (non-sterile) included inter-operative and final washing. After washing, the discs were dried in a Pol-Eko SLW 240 electric dryer (Pol-Eko, Wodzisław Śląski, Poland) at 105 ± 5 °C for a minimum of 60 min. After this stage, the products did not leave the packaging room. The next stage in the production of discs is to subject them to washing again in a disinfector washer. The discs were subjected to disinfectant washing with detergent in a closed circuit at 55 °C. The final rinse with thermal disinfection was conducted in demineralized water at 90 °C. Finally, drying was carried out at the same temperature for 40 min. After washing and drying, the discs were packed in a double layer comprising paper–foil bags. Each sample was packaged separately. The final step in preparing the discs was steam sterilization of the packed discs in an ASL 100MSV sterilizer, at 134 °C and an exposure time of 15 min.

#### MTT Test

Cell viability was assessed using the 3-(4,5-dimethylthiazol-2-yl)-2,5-diphenyltetrazolium bromide (MTT) assay, which takes advantage of the presence of mitochondrial succinate dehydrogenase only in living cells. Cells from 12-well plates were washed with 1 mL of PBS (0.02 M, pH 7.3). After pipetting off the buffer, 1 mL PBS and 25 µL MTT (5 mg MTT per 1 mL PBS) were added to the cells again. The plates were incubated for 10 min at 37 °C in a laboratory hothouse. Then, the liquid was pipetted off and the resulting crystals were dissolved in 1mL of DMSO solution (Invitrogen, D12345). After 10 min of incubation at room temperature, 10 µL of Sorensen buffer (0.1 mol/L glycine + 0.1 mol/L sodium chloride, pH 10.5) was added to the plates. The absorbance of the mixture was measured at 570 nm. The results for each group were expressed as a percentage of the control, which was considered 100%.

### 4.3. Biochemical Determinations

Total protein concentration was assessed in the mitochondrial suspension, as well as the activity of superoxide dismutase (SOD, EC 1.15.1.1), catalase (CAT, EC 1.11.1.6), glutathione peroxidase (GPx, EC 1.11.1.9), and the concentration of reduced glutathione (GSH). NADPH oxidase (NOX) activity, concentration of oxidative damage to lipids (4-hydroxynonenal-protein adducts (4-HNE)) and proteins (content of disulfide groups (SS), end products of advanced protein glycation (AGE)) were also assessed. Nitrosative stress parameters were evaluated: peroxynitrite (ONOO-) and 3-nitrotyrosine (3-NT). In addition, respiratory chain function was assessed by measuring the activities of complex I (EC 1.6.5.3) and complex II (EC 1.3.5.1), cytochrome c oxidase (EC 1.9.3.1, COX), and citrate synthase (EC 2.3.3.1, CS), as well as caspase-3 (CASP-3) activity. In the medium taken from the cells, the concentration of fibroblast growth factor FGF-2, the concentration of vascular endothelial growth factor (VEGF-A), and the content of metals (titanium (Ti), aluminum (AI), and vanadium (V)) were determined.

All assays were performed in duplicate. 96-well microplates were incubated in a DIS-4 Sky-Line shaker, Elmi, while Eppendorf-type tubes were incubated in a SalvisLab laboratory hothouse, (DanLab Białystok, Poland). ELISA microplate washing was performed using a Biotek 50 TS automatic microplate washer. Absorbance/fluorescence of the samples was measured using a Tecan Infinite M200 PRO Multimode microplate reader (Tecan Group Ltd., Männedorf, Switzerland). The results obtained were standardized per 1 mg of total protein.

Total protein content was determined by the bicinchoninic colorimetric method using the Thermo Scientific PIERCE BCA Protein Assay commercial diagnostic kit (Rockford, IL, USA). The composition of individual reagents is protected by the manufacturer’s secrecy.

The principle of the method is based on the production of a stable complex between a peptide band and bicinchoninic acid (BCA) and copper ions Cu^2+^, which shows a maximum of absorption at 562 nm.

### 4.4. Antioxidant Enzymes and Proteins

Catalase activity (CAT, EC 1.11.1.6) was measured colorimetrically using the Aebi method [57]. The principle of the method is based on measuring the decomposition rate of H_2_O_2_ at a wavelength of 240 nm. 1 unit of enzyme activity was defined as the amount of enzyme degrading 1 mol of H_2_O_2_ in 50 mM phosphate buffer pH 7.0 for 1 min at 25 °C. CAT activity was standardized to total protein and expressed in mol/min/mg of total protein.

Glutathione peroxidase (GPx, EC 1.11.1.9) activity was determined by a colorimetric method based on the reduction of glutathione with simultaneous oxidation of NADPH to NAD^+^ [58]. The amount of enzyme catalyzing the oxidation of 1 mol/L NADPH at 25 °C and pH 7.4 was defined as 1 unit of GPx activity. GPx activity was standardized to total protein content and expressed in mU/mg of total protein.

Superoxide dismutase (SOD, EC 1.19.1.1) activity was determined by a colorimetric method according to Misra et al. [59]. The principle of the method is to measure the activity of the cytoplasmic isoform of SOD during the inhibition reaction of the oxidation of epinephrine to adrenochrome. Absorbance was measured at a wavelength of 320 mm. 1 unit of SOD activity was assumed to inhibit epinephrine oxidation by 50% at 25 °C in 50 mM carbonate butter, and pH 10.2. SOD activity was standardized to total protein content and expressed in mU/mg of total protein.

Reduced glutathione (GSH) concentration was determined colorimetrically based on the reduction of 5,5-dithio-bis-(2-nitrobenzoic) acid (DTNB) to 2-nitro-5-mercaptobenzoic acid under the influence of GSH [60]. Absorbance was measured at 412 nm. GSH concentration was calculated from the standard curve for GSH solutions. GSH concentration was standardized to total protein and expressed in nM/mg of total protein.

### 4.5. Products of Oxidative Damage to Lipids and Proteins

The concentration of 4-hydroxynonenal adducts with proteins (4-HNE) was determined by immunoenzymatic ELISA using an off-the-shelf diagnostic kit (4-HNE Adduct Competitive ELISA, Cell Biolabs, San Diego, CA, USA). The composition of individual reagents is protected by manufacturer’s confidentiality. Absorbance was measured at 450 mm. The concentration of 4-HNE was calculated using a standard curve for HNE-BSA. The concentration of 4-HNE was standardized to total protein and expressed as nmol/mg of total protein.

The total level of disulfide groups (SS) was measured colorimetrically using Ellman’s reagent in 0.1 M phosphate buffer, pH 8.0. Absorbance was measured at 412 nm, and thiol group content was calculated from the standard curve with GSH as standard [61].

The content of advanced protein glycation end products (AGEs) was assessed using the fluorimetric method described by Kalousova et al. [62]. This method involves measuring the characteristic fluorescence of carbonyl derivatives of the AGE group, such as furoyl-furanyl-imidazole (FFI), carboxymethyllysine (CML), pyralin, and pentosidine. Measurements were made at an excitation wavelength of 350 nm and an emission wavelength of 440 nm.

AGE content was standardized to total protein content and expressed in arbitrary fluorescence units (AFU)/mg of total protein.

### 4.6. NADPH Oxidase Activity

The activity of NADPH oxidase (NOX, EC 1.6.3.1.) was determined using a chemiluminescent method. The principle of the method is to measure the rate of O_2_^•−^ formation in a reaction catalyzed by NOX using lucigenin as a luminophore. The amount of enzyme that causes the release of 1 mM O_2_^•−^ at 37 °C in 50 mM phosphate buffer pH 7.0 in 1 min was defined as 1 unit of NOX activity. NOX activity was standardized to total protein content and expressed as nM O_2_^•−^/min/mg protein total [7].

### 4.7. Nitrosative Stress

Peroxynitrite concentration (ONOO-) was determined by fluorimetric method [63] based on evaluation of the rate of nitration of phenol. The reaction of ONOO- with phenol produces p-nitrophenol, which shows an absorption maximum at an excitation wavelength of 490 nm and an emission wavelength of 530 nm. The fluorescence of the samples was measured at an excitation wavelength of 490 nm and an emission wavelength of 530 nm. To calculate the concentration of ONOO-, the molar absorption coefficient for p-nitrophenol ε = 1670 M^−1^ cm^−1^ was used. The concentration of ONOO- was standardized to total protein content and expressed in pmol/mg of total protein.

The concentration of 3-nitrotyrosine (3-NT) was determined by immunoenzymatic ELISA using a commercial diagnostic kit (Nitrotyrosin ELISA, Immundiagnostik, Ben sheina, Germany). The composition of the individual reagents is protected by manu facturer’s confidentiality. Absorbance was measured at 450 nm. The concentration of 3-NT was calculated using a standard curve for 3-NT. The concentration of 3-NT was standardized to total protein and expressed as pmol/mg of total protein.

### 4.8. Concentration of Growth Factors

FGF-2 and VEGF-A concentrations were determined by immunoenzymatic ELISA using the commercial ELISA Kit for Human Heparin-binding growth Factor 2 (HBGF-2, FGF-2) (EIAab, Wuhan, China) and ELISA Kit for IT man Vascular endothelial growth factor A (VEGF-A), ElAab, E0143h. The composition of each reagent is protected by manu facturer’s confidentiality. Absorbance was measured at 450 nm.

### 4.9. Mitochondrial Activity

The activity of mitochondrial complex I (EC 1.6.5.3) was determined by a colorimetric method based on the reduction of 2,6-dichloro-diphenol (DCIP) by electrons derived from decylubiquinol [64]. The decrease in absorbance at 600 nm was measured.

1 unit of mitochondrial complex I activity was defined as 1 µmol of DCIP reduced in 1 min at 37 °C. Mitochondrial complex I activity was standardized to total protein and expressed as mU/mg of total protein.

Mitochondrial complex II (EC 1.3.5.1) activity was determined by a colorimetric method based on measurement of succinate-ubiquinone reductase activity [64]. The decrease in absorbance at 600 m was measured. Mitochondrial complex II activity was standardized to total protein content and expressed as mU/mg of total protein.

Cytochrome c oxidase (EC 1.9.3.1, COX) activity was determined by a colorimetric method by measuring the oxidation of reduced cytochrome c at 550 nm [65]. Cytochrome c oxidase activity was standardized to total protein and expressed as mU/mg of total protein.

Citrate synthase (EC 2.3.3.1, CS) activity was assessed colorimetrically using a method with 5-thio-2-nitrobenzoic acid, which is formed from 5,5′-dithiobis-2-nitrobenzoic acid during the CS synthesis reaction [65]. Absorbance was measured at 340 nm. CS activity was standardized to total protein content and expressed as mU/mg of total protein.

Caspase-3 (CASP-3) activity was assessed by colorimetric assay using Ac-Asp-Glu-Val-Asp-p-nitroanilide as substrate [66]. The principle of the method is based on the reaction catalyzed by CASP-3, which will release p-nitroaniline (pNA) from the substrate. The maximum absorption of pNA occurs at 405 nm. CASP-3 activity was standardized to total protein and expressed in μmol/min/mg of total protein.

### 4.10. Determination of Metal Content

#### 4.10.1. Sample Preparation

In order to determine the metal content of the samples, cell medium was taken every 3 days for 3 weeks (the first medium was taken on the 3rd day after the titanium plates were applied), in which aluminum, titanium, and vanadium were determined by ICP-MS (inductively coupled plasma mass spectrometry). The medium from above the cells and the control medium were mineralized in the same way. In a 12-mL centrifuge test tube (Deltalab), 3 mL of cell medium was transferred and 0.65 mL of 65% nitric acid (HNO_3_) and 0.35 mL of 30% H_2_O_2_ were added. The test tubes were capped with a stopper and placed in a water bath at 90 °C for 90 min. The samples were then allowed to cool and degassed in an ultrasonic bath for 30 min (100% power). After degassing, the samples were diluted with deionized water to a volume of 9 mL and stored at 4 °C until analysis. To control the accuracy of the measurements, a recovery experiment was performed. For this purpose, the medium samples were enriched with Al, Ti and V standards at three different concentrations: 3 ng mL^−1^, 5 ng mL^−1^, and 6 ng mL^−1^ (n = 3), and then digested in a manner analogous to the medium samples from the cells.

#### 4.10.2. Measuring Apparatus

Concentrations of aluminum, titanium, and vanadium in solutions after mineralization of the cell medium were measured by inductively coupled plasma ionization mass spectrometry ICP-MS. For this purpose, an 8800 Triple Quad ICP-MS spectrometer (Agilent Technologies, Singapore) equipped with an SPS4 sample feeder, a MicroMist nebulizer, a Scott-type mist chamber cooled by a Peltier system, nickel sampler and collector cones, and an ORS^3^ reaction–collision chamber was used. In order to remove spectral interference during the determination of metals, helium was used as a collision gas and ammonia as a reaction gas in the ORS^3^ chamber. Each measurement day, standard tuning of the ICP-MS spectrometer was carried out to control its proper operation. Agilent Mass Hunter software was used to collect and process measurement data.

#### 4.10.3. Quantitative Analysis

Quantitative determinations in mineralized samples of cell medium were conducted using the calibration curve method. To prepare standard solutions of analytes: Al, Ti, and V, single element standard solutions were used, from which working standards were prepared by appropriate dilution. Calibration curves of Al, Ti, and V were made in the concentration range from 0.5 to 50 ng mL^−1^ in the control medium solution after mineralization. A Rh solution of 100 ng mL^−1^ was used as an internal standard to compensate for the influence of the matrix.

Instrument analyte limits of detection (LOD), calculated as 3SD_blank_/a (a-slope of the calibration curve), were, respectively: 0.645 ng mL^−1^ for Al, 0.417 ng mL^−1^ for Ti, and 0.052 mg mL^−1^ for V. The accuracy of the method was confirmed each measurement day in a recovery experiment by analysing mineralized medium samples enriched with analytes. Metal recoveries were 102–106% for Al, 101–106% for Ti, and 91–102% for V.

#### 4.10.4. Statistical Analysis

Statistical analyses were performed using GraphPad Prism 9.0 software (GraphPad Software, San Diego, CA, USA). Normality of distribution was checked using the Shapiro–Wilko test, while homogeneity of variance was checked using the Brown–Forsythe test. Quantitative variables were described by parameters of descriptive statistics, i.e., arithmetic mean and standard deviation. One-way ANOVA analysis of variance with Tukey’s HSD post-hoc test was used for group comparisons. A significance level of less than 0.05 was assumed for the statistical analyses performed.

Person’s parametric correlation test was used to assess the relationship between quantitative variables.

## 5. Conclusions

The II type of anodization of the titanium surface induced slight difference in the antioxidant response of human, non-genetically modified, primary gingival fibroblasts exposed to titanium discs;

Type II anodization: prevented changes in complex II activity (vs. control); inhibited CS activity the least, as well as in the final phase of culture, reduced the degree of COX inhibition and prevented apoptosis in the mitochondria compared to the other test groups and the control group;

The obtained results proved the existence of mitochondrial dysfunction and redox imbalance in the mitochondria of fibroblasts exposed to hard-anodized titanium discs, which suggests the need to search for new materials perhaps biodegradable in body tissues.

## Figures and Tables

**Figure 1 ijms-24-12896-f001:**
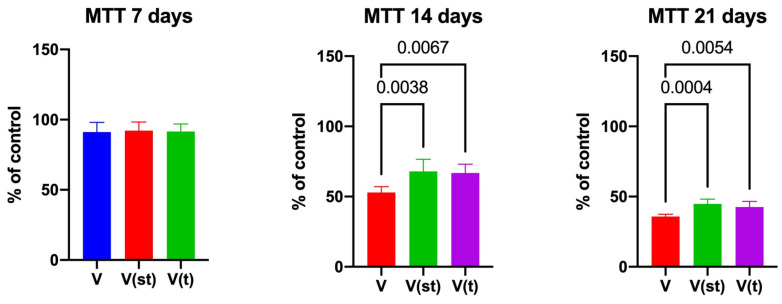
MTT test of the fibroblasts exposed to a titanium alloy with an admixture of vanadium. V—Ti6Al4V alloy without coating, V(st)—Ti6Al4V alloy with standard anodized coating, and V(t)—Ti6Al4V alloy hard anodized. One-way ANOVA analysis of variance with Tukey’s HSD post-hoc test was used and results are presented as mean ± SD.

**Figure 2 ijms-24-12896-f002:**
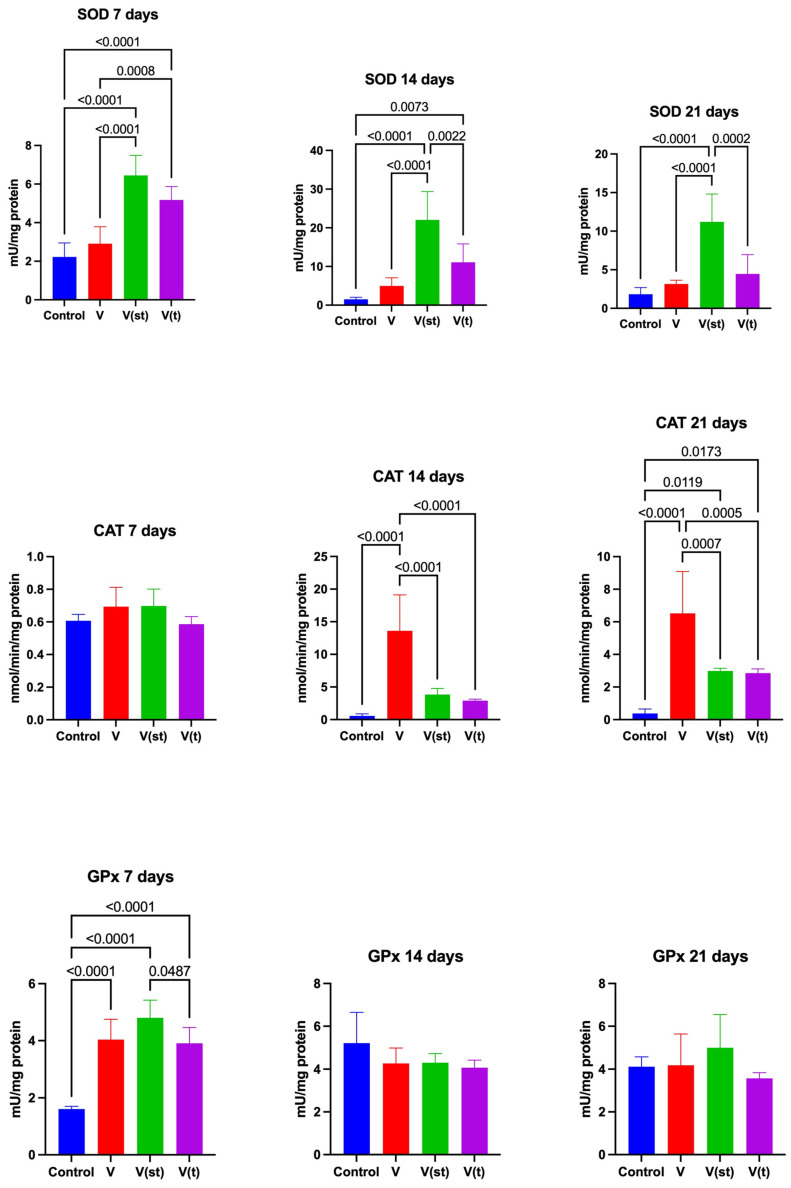
Mitochondrial antioxidant enzymes in the fibroblasts exposed to a titanium alloy with an admixture of vanadium. V—Ti6Al4V alloy without coating, V(st)—Ti6Al4V alloy with standard anodized coating, V(t)—Ti6Al4V alloy hard anodized, SOD—superoxide dismutase, CAT—catalase, and GPx—Glutathione peroxidase. One-way ANOVA analysis of variance with Tukey’s HSD post-hoc test was used and results are presented as mean ± SD.

**Figure 3 ijms-24-12896-f003:**
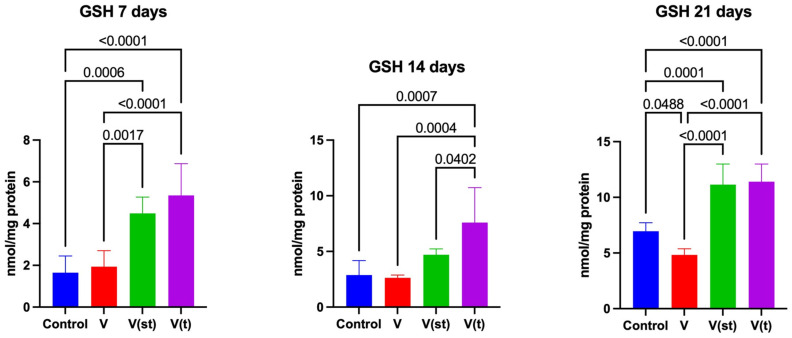
Mitochondrial reduced glutathione in the fibroblasts exposed to a titanium alloy with an admixture of vanadium. V—Ti6Al4V alloy without coating, V(st)—Ti6Al4V alloy with standard anodized coating, V(t)—Ti6Al4V alloy hard anodized, and GSH—reduced glutathione. One-way ANOVA analysis of variance with Tukey’s HSD post-hoc test was used and results are presented as mean ± SD.

**Figure 4 ijms-24-12896-f004:**
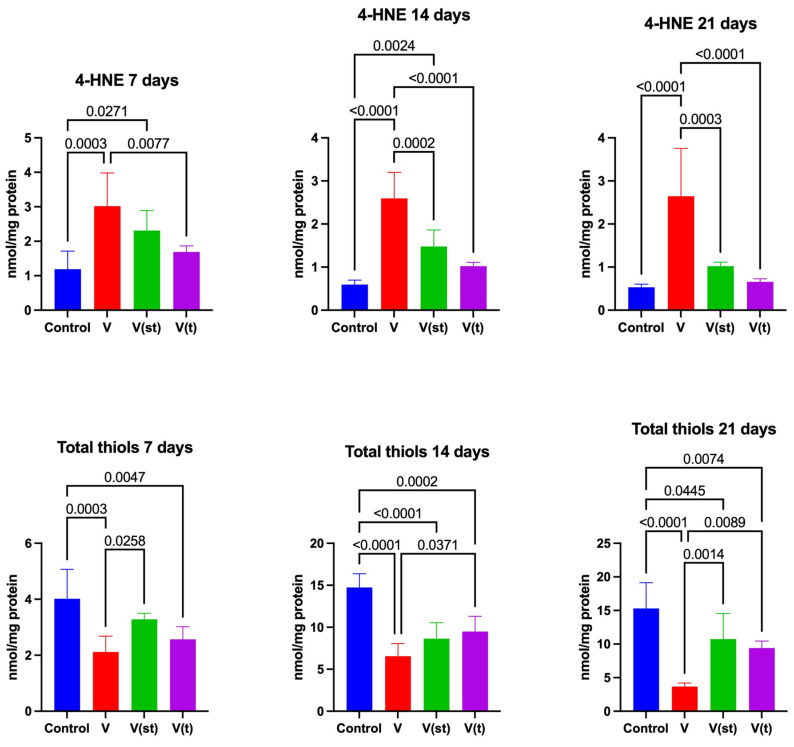
Mitochondrial oxidation products in the fibroblasts exposed to a titanium alloy with an admixture of vanadium. V—Ti6Al4V alloy without coating, V(st)—Ti6Al4V alloy with standard anodized coating, V(t)—Ti6Al4V alloy hard anodized, 4—HNE adducts of 4 hydroxynonenal, and AGE—advanced protein glycation end products. One-way ANOVA analysis of variance with Tukey’s HSD post-hoc test was used and results are presented as mean ± SD.

**Figure 5 ijms-24-12896-f005:**
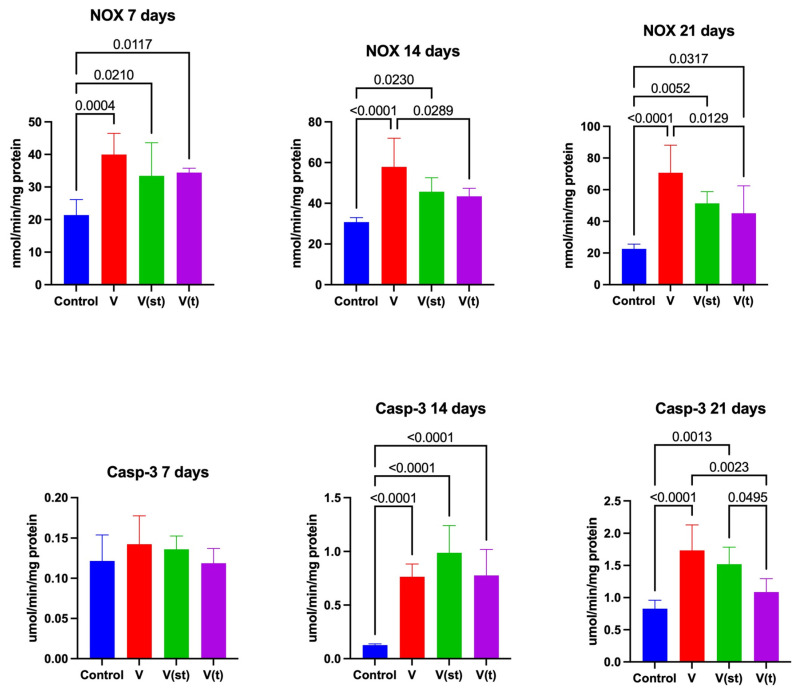
ROS production and apoptosis in the fibroblasts exposed to a titanium alloy with an admixture of vanadium. V—Ti6Al4V alloy without coating, V(st)—Ti6Al4V alloy with standard anodized coating, V(t)—Ti6Al4V alloy hard anodized, NOX—oxidase NADPH, and Casp-3—caspase 3. One-way ANOVA analysis of variance with Tukey’s HSD post-hoc test was used and results are presented as mean ± SD.

**Figure 6 ijms-24-12896-f006:**
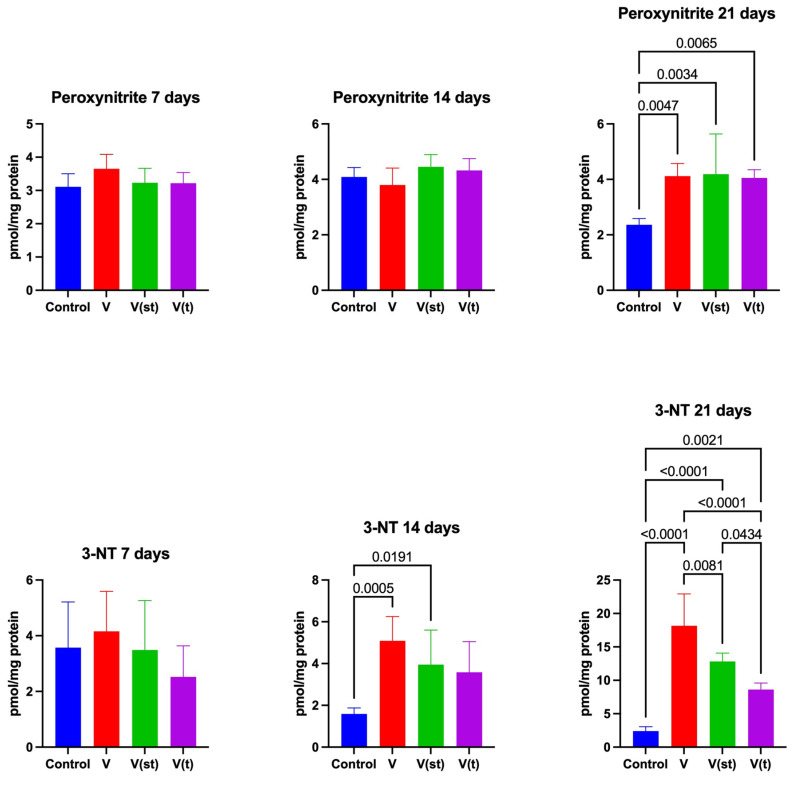
Mitochondrial nitrosative stress in the fibroblasts exposed to a titanium alloy with an admixture of vanadium. V—Ti6Al4V alloy without coating, V(st)—Ti6Al4V alloy with standard anodized coating, V(t)—Ti6Al4V alloy hard anodized, and 3-NT—3-nitrotyrosine. One-way ANOVA analysis of variance with Tukey’s HSD post-hoc test was used and results are presented as mean ± SD.

**Figure 7 ijms-24-12896-f007:**
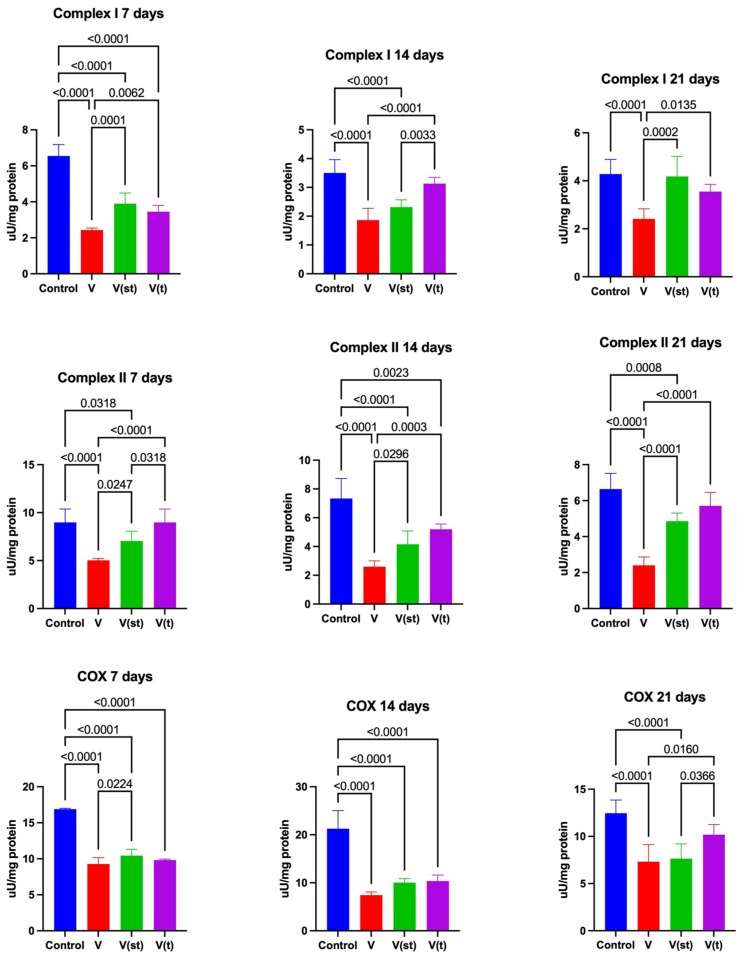
Mitochondrial complexes in the fibroblasts exposed to a titanium alloy with an admixture of vanadium. V—Ti6Al4V alloy without coating, V(st)—Ti6Al4V alloy with standard anodized coating, V(t)—Ti6Al4V alloy hard anodized, COX—cytochrome c oxidase, and CS—citrate synthase. One-way ANOVA analysis of variance with Tukey’s HSD post-hoc test was used and results are presented as mean ± SD.

**Figure 8 ijms-24-12896-f008:**
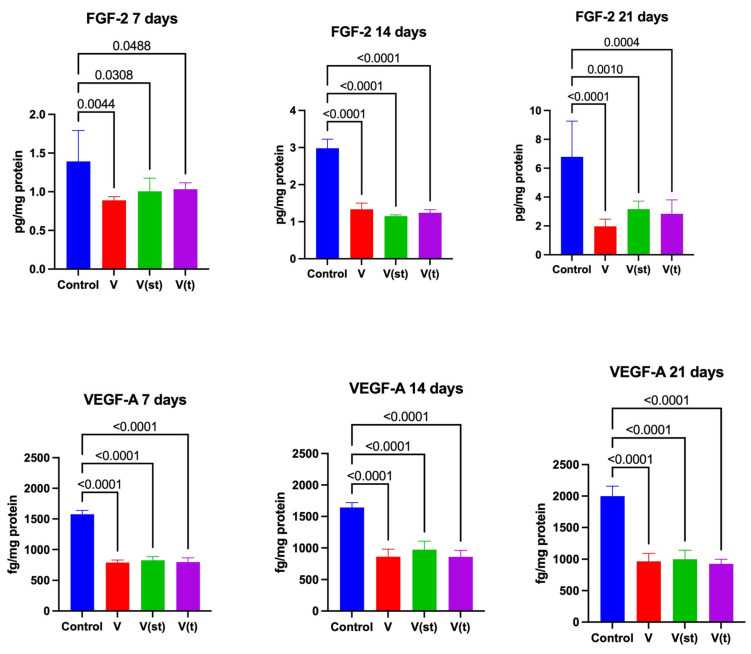
Growth in the medium collected from the fibroblasts exposed to a titanium alloy with an admixture of vanadium. V—Ti6Al4V alloy without coating, V(st)—Ti6Al4V alloy with standard anodized coating, V(t)—Ti6Al4V alloy hard anodized, FGF-2—fibroblast growth factor 2, and VEGF-A—vascular endothelial growth factor A. One-way ANOVA analysis of variance with Tukey’s HSD post-hoc test was used and results are presented as mean ± SD.

**Table 1 ijms-24-12896-t001:** Assessment of the titanium content in the medium collected from the fibroblasts exposed to a titanium alloy with an admixture of vanadium.

	V (ng mL^−1^)	V(st) (ng mL^−1^)	V(t) (ng mL^−1^)
3rd day	55.45 ± 1.35 **	53.45 ± 2.48 ***	115.59 ± 4.30
6th day	42.74 ± 0.55 *^,^**	27.97 ± 1.34 ***	124.53 ± 5.03
15th day	19.28 ± 0.28 *^,^**	12.68 ± 0.06 ***	63.53 ± 2.07
21st day	18.00 ± 0.58 *^,^**	10.78 ± 0.44 ***	63.65 ± 0.79

V—Ti6Al4V alloy without coating, V(st)—Ti6Al4V alloy with standard anodized coating, V(t)—Ti6Al4V alloy hard anodized. * V vs. V(st); ** V vs. V(t); *** V(st) vs. V(t). One-way ANOVA analysis of variance with Tukey’s HSD post-hoc test was used and results are presented as mean ± SD.

**Table 2 ijms-24-12896-t002:** Evaluation of the aluminum content in the medium collected from over the fibroblasts exposed to a titanium alloy with an admixture of vanadium.

	V (ng mL^−1^)	V(st) (ng mL^−1^)	V(t) (ng mL^−1^)
3rd day	12.00 ± 0.42 **	13.53 ± 0.55 ***	39.87 ± 2.12
6th day	10.42 ± 0.37 *^,^**	4.15 ± 0.10 ***	49.50 ± 2.44
15th day	28.61 ± 0.75 **	1.37 ± 0.03 ***	31.04 ± 2.17
21st day	5.08 ± 0.16 ***	0 ± 0	21.40 ± 0.99

V—Ti6Al4V alloy without coating, V(st)—Ti6Al4V alloy with standard anodized coating, and V(t)—Ti6Al4V alloy hard anodized. * V vs. V(st); ** V vs. V(t); *** V(st) vs. V(t). One-way ANOVA analysis of variance with Tukey’s HSD post-hoc test was used and results are presented as mean ± SD.

**Table 3 ijms-24-12896-t003:** Evaluation of the vanadium content in the medium collected from the fibroblasts exposed to a titanium alloy with an admixture of vanadium.

	V (ng mL^−1^)	V(st) (ng mL^−1^)	V(t) (ng mL^−1^)
3rd day	9.09 ± 0.31 **	10.73 ± 0.42 ***	190.40 ± 3.30
6th day	2.09 ± 0.07 **	1.67 ± 0.01 ***	97.96 ± 0.44
15th day	0.68 ± 0.02 *^,^**	0.27 ± 0.01 ***	37.79 ± 0.77
21st day	0.87 ± 0.03 **	0.79 ± 0.04 ***	25.99 ± 0.84

V—Ti6Al4V alloy without coating, V(st)—Ti6Al4V alloy with standard anodized coating, and V(t)—Ti6Al4V alloy hard anodized. * V vs. V(st); ** V vs. V(t); *** V(st) vs. V(t). One-way ANOVA analysis of variance with Tukey’s HSD post-hoc test was used and results are presented as mean ± SD.

## Data Availability

All of the data used to support the findings of this study are included within the article.

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
