# Peer review of "Mitochondrial Redox Balance of Fibroblasts Exposed to Ti-6Al-4V Microplates Subjected to Different Types of Anodizing"

_ijms, 2023, doi:10.3390/ijms241612896_

Round 1

Reviewer 1 Report

Well written and engaging work. All issues regarding the impact of implants on the human body are most valuable. My congratulations, well done extensively conducted experiments. However as a reviewer I have a few minor remarks.

You use a lot of abbreviations, I know that you are experts in this area, but for an engineer who deals with the preparation of the surface itself, some of these markings are unknown. I would suggest at the end of the work to add all the abbreviations with their explanations.

  thank you

material and methods

  Please specify the manufacturers of all reagents used for the tests and the country of manufacture. The same applies to the equipment, e.g. microscope and magnification used for testing.

you used cells of human origin, I would like permission from your university to use this type of material

Line 696

, 20 mM HEPES- please explain this aberration

Line 706

Were the titanium alloy discs made by you, if so, how? and if you get it, give the source.

Line 710

All discs were polished according to a standard polishing procedure to a surface roughness equal to Ra0.63.- please provide the ref for this treatment.  What kied of polishing paste was used for this?

Line 720

Anodic Treatment - Titanium and Titanium Alloys Solution pH or Higher- please provide the reference

Line 726

n aqueous detergent solution- producer, country? 

MTT test

  This device was used for testing at 570 nkm. Was a blind test used? This may pose a problem in interpreting the results. Because if there was no reference test, what was the survival rate of the cells themselves after 7.14, 21 days?

Line 914

 Graphpad Prism 9.0 software- producer country, please

Results

Figure 1. You state that the titanium alloy was modified with Vanadium and there is no word about it in materials and methods. It would be worth supplementing.

Table 1, 2,3 There are no units in which the concentration of eluting metals was specified?

Good luck with your future research!

Author Response

Well written and engaging work. All issues regarding the impact of implants on the human body are most valuable. My congratulations, well done extensively conducted experiments. However as a reviewer I have a few minor remarks.

You use a lot of abbreviations, I know that you are experts in this area, but for an engineer who deals with the preparation of the surface itself, some of these markings are unknown. I would suggest at the end of the work to add all the abbreviations with their explanations.

The abbreviation were added at the end of the paper.

  thank you

 material and methods

  Please specify the manufacturers of all reagents used for the tests and the country of manufacture. The same applies to the equipment, e.g. microscope and magnification used for testing

We specified the manufacturers of all reagents used for the tests and the country of manufacture, all changes in the red color throughout the text.

you used cells of human origin, I would like permission from your university to use this type of material

The study was approved by the Bioethics Committee of the Medical University of Bialystok (resolution no. APK.002.280.2021).

Line 696

20 mM HEPES- please explain this aberration

20 mM HEPES (hydroxyethyl piperazine ethanesulfonic acid, Sigma Aldrich, H4034)

Line 706

Were the titanium alloy discs made by you, if so, how? and if you get it, give the source.

The discs were made to individual order by ChM sp z o.o. (Lewickie, Poland).

Line 710

All discs were polished according to a standard polishing procedure to a surface roughness equal to⩽ Ra0.63.- please provide the ref for this treatment.  What kied of polishing paste was used for this?

All discs were polished according to a standard polishing procedure to a surface roughness of Ra⩽0.63. Rosler Keramo-Finish Compound 629 (LM Zalewski, Poland) was used for polishing.

Line 720

Anodic Treatment - Titanium and Titanium Alloys Solution pH or Higher- please provide the reference

Type II (hard anodizing): the disks were anodized in accordance with AMS

2488D ("Anodic Treatment - Titanium and Titanium Alloys Solution pH 13 or Higher"), this specification describes the technical requirements related to the production of an electrolytic conversion (anodic) coating on titanium and titanium alloys. It describes the properties of the coating.” https://webstore.ansi.org/standards/sae/saeams2488d2000ams2488d. https://webstore.ansi.org/standards/sae/saeams2488d2000ams2488d.

Line 726

aqueous detergent solution- producer, country?

After the treatment process, the discs were rinsed in water and then full immersed in an aqueous detergent solution (Eskaphor N6650, Haugh Chemie, Poland) 

MTT test

  This device was used for testing at 570 nkm. Was a blind test used? This may pose a problem in interpreting the results. Because if there was no reference test, what was the survival rate of the cells themselves after 7.14, 21 days?

As with the other assays, cell survival was also determined in four groups: C, V, V(st) and V(t). Following the conventional way of presenting results, MTT results were presented as a percentage of control (C) - that is why there are only 3 bars on each graph.

Line 914

 Graphpad Prism 9.0 software- producer country, please

Statistical analyses were performed using Graphpad Prism 9.0 (GraphPad Software, San Diego, California, USA).

Results

Figure 1. You state that the titanium alloy was modified with Vanadium and there is no word about it in materials and methods. It would be worth supplementing.

The study used discs with a diameter of 21 mm and a thickness of 1 mm made of titanium alloy Ti-6AI-4V containing a minimum of 88 wt % titanium, as well as 6 wt % aluminium and 4 wt % vanadium. Other titanium alloys produced by ChM do not contain vanadium. They may contain an admixture of niobium or a higher carbon content.

Table 1, 2,3 There are no units in which the concentration of eluting metals was specified?

The units were added in the tabels

Reviewer 2 Report

1.       I would recommend the authors to extent alternative surface modifications to provide the authors more diverse vision of the field, e..g reported by Douglas et al doi.org/10.3390/ijms21176406; doi.org/10.3390/mi10010068

2.       The authors provide extensive biological characterization with detailed analysis of the results obtained. However, in respect with the chemical and structural features of the developed materials no detailed have been discussed. At least I would recommend the authors to provide the most important details on the substrates and the properties after surface anodization.

3.       Anodization results in surface porosity. Did the authors observed any porous. If so. Provide dimensions and other quantitative details.

4.       The authors report the release of ions for example V, Al etc. Provide some qualitative and quantitative contents, which define the outcome of biological trials.

5.       It is reported in the literature that ROS content to a certain extent may provide positive and even stimulating effect on cells. Could the authors provide some quantitative data on ROS values which might have affected the results obtained?

Author Response

I would recommend the authors to extent alternative surface modifications to provide the authors more diverse vision of the field, e..g reported by Douglas et al doi.org/10.3390/ijms21176406doi.org/10.3390/mi10010068

These sentences were added

In the case of biomaterials such as titanium alloys used in bone reconstruction, the promotion of osteogenic differentiation and reduction of inflammation and thus OS are desirable. That is why researchers are looking for various solutions to increase the biocompatibility of titanium implants. Mieszkowska A et all [13] use fibrillar coatings with phloroglucinol (PG). These studies have shown that this type of coatings significantly reduces the inflammatory response as well as promotes osteogenic differentiation. In the studies of Norris K et all [14], collagen fibril sheaths enriched with HE800 exopolysaccharides and GY785 EPS heteropolysaccharide derivatives were deposited on a titanium alloy (Ti6Al4V). These envelopes did not significantly alter the adhesion, morphology and viability of osteoblast-like cells. In these studies, the influence of coatings on the behaviour of oxidative stress (OS) parameters was not assessed.

To reduce harmful influence of titanium discs on redox balance and, consequently, on implant disintegration could be the introduction of titanium implants with a thickened titanium dioxide layer.

2.The authors provide extensive biological characterization with detailed analysis of the results obtained. However, in respect with the chemical and structural features of the developed materials no detailed have been discussed. At least I would recommend the authors to provide the most important details on the substrates and the properties after surface anodization. The metallographic microsection of the Ti6Al4V sample was analysed using the FEI Quanta 3D FEGSEM scanning electron microscope equipped with an EDAX X-ray spectrometer.

Observations were carried out using backscattered electron (BSE) images at 2000x - 10000x magnification. Analysis of chemical composition - X-ray spectroscopy with energy dispersion. The thickness of the layer on the exterminations measured in an electron microscope was in the range of 1-2 micrometres; the average value was 1.5 +/- 0.5). It was not technically possible to prepare microsections for measuring the thickness of the standard - layer thickness was too small. The images obtained suggest that the surface topography of both types of layers was similar. Coating nanohardness tests determined by the instrumental method on the device Micro Combi Tester – Anton Paar. The nanohardness value of the type II anodic layer was 160% of the value of the standard layer.

3.Anodization results in surface porosity. Did the authors observed any porous. If so. Provide dimensions and other quantitative details.

The discs were prepared by the ChM Lewickie company, where they were carefully checked as I mentioned above. No pores were observed in the structure of the titanium discs. The discs were used once.

4.The authors report the release of ions for example V, Al etc. Provide some qualitative and quantitative contents, which define the outcome of biological trials.

I'm sorry but I don't understand this Reviewer's request. Please specify what we would like to describe. Thank you in advance!

5.It is reported in the literature that ROS content to a certain extent may provide positive and even stimulating effect on cells. Could the authors provide some quantitative data on ROS values which might have affected the results obtained?

On the other hand, it is known that high concentrations of H2O2 may be desirable in the process of indentation of a titanium implant and in wound healing in general. The generation of large amounts of H2O2 at the implant/wound site and the resulting gradient in its concentration are essential for rapid leukocyte recruitment and thus implant engraftment/wound healing.

Low-intensity OS can have a very beneficial effect, as it increases cell survival. This effect is associated with the activation of a number of cellular processes directed at increasing cellular resistance to OS. The reversibility of lipid peroxidation is of great clinical importance. Lipid peroxidation products are considered a marker of osteoclast activity, and it is even believed that their concentration is directly proportional to osteoclast activity around titanium implants and screws.

However, it should be remembered that exogenous antioxidants used for bone healing are beneficial in patients with systemic diseases such as diabetes and osteoporosis. The effect of these antioxidants on bone healing in healthy patients seems to be different. In the latter case, the antioxidants are believed to neutralize the levels of ROS necessary for the healing process. Physiological levels of ROS play a role of second messengers to mediate responses via redox reactions, that are required for proper cell function and survival.

Reviewer 3 Report

The presented manuscript by Zalewska et al. focuses on a microplates subjected to a novel type of anodizing and their suitability for implantation. The general subject of the study seems interesting, however, the Authors presented a huge number of results that, in fact, were not supporting their hypothesis concerning higher biocompatibility of "hard anodization" process.

Major remarks:

1/ I would suggest the Authors to re-write their manuscript, formulate the clear aim of the study, not focused on the biocompatibility, but rather on the harmful influence of titanium discs on redox balance

2/ please explain why so many parameters were evaluated in isolated mitochondria, not in whole cells? Is it reliable? Oxidative damage to lipids and proteins occurs primarily at the cell membrane, why was it not determined?

3/ please explain the validity of FGF level determination in culture media - this growth factor is added to the medium as supplement, so changes in the cellular production of FGF will be probably masked.

4/ Introduction could be completed with information on maintaining the redox balance in cells

5/ the Discussion is very long and hard to follow, some of the information could be placed in Introduction

6/ In the Conclusions section the Authors must clearly formulate their main achievements, and what do they want to say with this set of data

Minor remarks:

1/ Figure 1 - why the viability results on day 7 lack V(t) sample?

2/ descriptions of all the figures could be more informative, with full names of parameters tested, and description of statistical evaluation.

3/ why are "Correlations" divided as a separate subchapter? and what does it mean - that the only correclation found was this one described there?

4/ some sentences in Abstract are in present time, some in past - -please correct it

5/ line 24 - mitochondrial fibroblasts - correct to "fibroblasts mitochondria"

6/ line 36 - redox balance - or rather "redox imbalance"?

7/ line 378 - "MTT assay" instead of MIT assay

8/ line 414 - "oxidative burst" instead of "oxygen explosion"

The English is decent, some improvements in grammar or wording are required.

Author Response

The presented manuscript by Zalewska et al. focuses on a microplates subjected to a novel type of anodizing and their suitability for implantation. The general subject of the study seems interesting, however, the Authors presented a huge number of results that, in fact, were not supporting their hypothesis concerning higher biocompatibility of "hard anodization" process.

Major remarks:

1/ I would suggest the Authors to re-write their manuscript, formulate the clear aim of the study, not focused on the biocompatibility, but rather on the harmful influence of titanium discs on redox balance

The text of the paper was re-written.

2/ please explain why so many parameters were evaluated in isolated mitochondria, not in whole cells? Is it reliable? Oxidative damage to lipids and proteins occurs primarily at the cell membrane, why was it not determined?

Mitochondria are the main source of ROS and thus the main cause of redox imbalances. Therefore, we selected them for research on the redox balance in fibroblast cells exposed to titanium discs.

It is true that lipids and membrane proteins are also subject to oxidation processes, but this is material for a separate publication.

The number of determinations is large, but they allow for a precise characterization of the processes taking place in the mitochondria of fibroblasts exposed to titanium discs. A single biomarker of oxidative damage having little clinical value in diagnostic or prognostic terms.

3/ please explain the validity of FGF level determination in culture media - this growth factor is added to the medium as supplement, so changes in the cellular production of FGF will be probably masked.

It is true that the medium was supplemented with growth factor FGF-2. At the same time, each sample was treated similarly, so the observed changes result from disturbances in the functioning of cells exposed to the titanium discs.

4/ Introduction could be completed with information on maintaining the redox balance in cells

Physiologically, a living organism maintains a balance between the production and inactivation of ROS, which is called redox balance. The generation of ROS in cells using oxygen as an energy source is coupled with the existence of protective systems against their action, the so-called antioxidant barrier of the body. One of the key antioxidant systems of cells is the SOD, CAT and GPx system. GPx, along with catalase, neutralizes H2O2 formed in a dismutation reaction involving SOD. The dismutation reaction, is the dismutation reaction of O2•- to H2O2 This includes NOX, which generates large amounts of O2•-, and sometimes H2O2 [8].

5/ the Discussion is very long and hard to follow, some of the information could be placed in Introduction

The discussion section was shortened and as Reviewer suggested some of the information was placed in introduction

6/ In the Conclusions section the Authors must clearly formulate their main achievements, and what do they want to say with this set of data

The conclusion were changed

  1. The II type of anodization of the titanium surface induced slight difference in the antioxidant response of human, non-genetically modified, primary gingival fibroblasts exposed to titanium discs.
  2. Type II anodization: prevented changes in complex II activity ( control); inhibited CS activity the least, as well as in the final phase of culture, reduced the degree of COX inhibition and prevented apoptosis in the mitochondria compared to the other test groups and the control group.
  3. The obtained results proved the existence of mitochondrial dysfunction and redox imbalance in the mitochondria of fibroblasts exposed to hard-anodized titanium discs, which suggests the need to search for new materials perhaps biodegradable in body tissues.

Minor remarks:

1/ Figure 1 - why the viability results on day 7 lack V(t) sample?

The Figure was corrected

2/ descriptions of all the figures could be more informative, with full names of parameters tested, and description of statistical evaluation.

The descriptions of all the Figures were corrected according to Reviewer’s suggestions.

3/ why are "Correlations" divided as a separate subchapter? and what does it mean - that the only correclation found was this one described there?

Correlations have been separated as a separate subsection for better visibility. Of the correlations we tested, this one was statistically significant.

4/ some sentences in Abstract are in present time, some in past - -please correct it

This point was addressed.

5/ line 24 - mitochondrial fibroblasts - correct to "fibroblasts mitochondria"

This point was addressed.

6/ line 36 - redox balance - or rather "redox imbalance"?

This point was addressed.

7/ line 378 - "MTT assay" instead of MIT assay

This point was addressed.

8/ line 414 - "oxidative burst" instead of "oxygen explosion"

This point was addressed.

Comments on the Quality of English Language

The English is decent, some improvements in grammar or wording are required.

The paper was editing by native English speaker.

Reviewer 4 Report

The manuscript was dedicated to a study entitled "Mitochondrial redox balance of fibroblasts exposed to Ti-6Al-4V microplates subjected to different types of anodizing". The manuscript itself is theoretical and structurally relevant. Also, it is original and results are generally very well supported by experimental evidence. I recommend this paper may be considered for publication after addressing the following points:

- There are minor grammatical errors throughout the manuscript. Bear in mind the use of past tense and present tense. Use past tense when referring to experiments but present tense when referring to scientific law, previously established knowledge, as well as a description of parameters.

- Please revise the typo errors. Authors should place the full stop or comma after the reference number in the bracket [ ], not before it.

- The negative effects of releasing vanadium and aluminum from Ti64 on the implantation site should be mentioned in the introduction section.

- On page 15, line 405, the sentence “The cathodic part of the corrosion process involves oxygen reduction at physiological pH with the generation of ROS and non-radical compounds, such as hydrogen peroxide H2O2” needs also the following references: https://doi.org/10.1038/s41598-023-29553-5; https://doi.org/10.3390/molecules28124837.

- Anodic layers’ thickness – has not been mentioned in the manuscript. Is this property important for the release of ions during inflammation?

- “Ti-6AI-4V containing a minimum of 88% titanium, as well as 6% aluminum 704 and 4% vanadium”. The values are in wt.% or at.%? 

- The conclusion section contains too much content, indicating that the focus of the author's review is not focused enough.

Minor editing of English language required!

Author Response

The manuscript was dedicated to a study entitled "Mitochondrial redox balance of fibroblasts exposed to Ti-6Al-4V microplates subjected to different types of anodizing". The manuscript itself is theoretical and structurally relevant. Also, it is original and results are generally very well supported by experimental evidence. I recommend this paper may be considered for publication after addressing the following points:

- There are minor grammatical errors throughout the manuscript. Bear in mind the use of past tense and present tense. Use past tense when referring to experiments but present tense when referring to scientific law, previously established knowledge, as well as a description of parameters.

This point was corrected, all the changes are in red throughout the text.

- Please revise the typo errors. Authors should place the full stop or comma after the reference number in the bracket [ ], not before it.

This point was corrected, all the changes are in red throughout the text.

- The negative effects of releasing vanadium and aluminum from Ti64 on the implantation site The negative effects of releasing vanadium and aluminum from Ti64 on the implantation site should be mentioned in the introduction section.

It was proven, that titanium implant’s aluminum (Al) and vanadium (V) constituens have very cytotoxic effect on cells and can be released for a long time from implantation, resulting harmful biological effects [6].

- On page 15, line 405, the sentence “The cathodic part of the corrosion process involves oxygen reduction at physiological pH with the generation of ROS and non-radical compounds, such as hydrogen peroxide H2O2” needs also the following references: https://doi.org/10.1038/s41598-023-29553-5; https://doi.org/10.3390/molecules28124837.

These references were added.

- Anodic layers’ thickness – has not been mentioned in the manuscript. Is this property important for the release of ions during inflammation?

- “Ti-6AI-4V containing a minimum of 88% titanium, as well as 6% aluminum 704 and 4% vanadium”. The values are in wt.% or at.%? 

The study used discs with a diameter of 21 mm and a thickness of 1 mm made of titanium alloy Ti-6AI-4V containing a minimum of 88 wt % titanium, as well as 6 wt % aluminium and 4 wt % vanadium. Other titanium alloys produced by ChM do not contain vanadium. They may contain an admixture of niobium or a higher carbon content. The titanium discs were made in three varieties depending on the process of formation of the passive layer: standard anodized discs (III type) (V(st)), cold anodized discs (II type) (V(t) the thickness of the layer on the exterminations measured in an electron microscope was in the range of 1-2 micrometers; the average value was 1.5 +/- 0.5)

We used one thickness of the anode layer, it is not known how other thicknesses of this layer would affect biological systems. It is also not known whether the release of ions would be at a similar level.

- The conclusion section contains too much content, indicating that the focus of the author's review is not focused enough.

The conclusion were changed

  1. The II type of anodization of the titanium surface induced slight difference in the antioxidant response of human, non-genetically modified, primary gingival fibroblasts exposed to titanium discs.
  2. Type II anodization: prevented changes in complex II activity ( control); inhibited CS activity the least, as well as in the final phase of culture, reduced the degree of COX inhibition and prevented apoptosis in the mitochondria compared to the other test groups and the control group.
  3. The obtained results proved the existence of mitochondrial dysfunction and redox imbalance in the mitochondria of fibroblasts exposed to hard-anodized titanium discs, which suggests the need to search for new materials perhaps biodegradable in body tissues.

Round 2

Reviewer 2 Report

can be accepted

Reviewer 3 Report

The Authors have replied to my comments and improved the indicated parts of the manuscript. 

Reviewer 4 Report

The authors justified the queries made by this reviewer. The revised version may be accepted in its current form.

Some typo errors still exist. Please revise them!